# Quadratically Regularized Optimal Transport: Localization Bounds and Affine Case Analysis

**Long Nguyen-Chi** [1]  **Nam Nguyen** [1]  **Binh T. Nguyen** [1 2]

## Abstract

Quadratic regularization has emerged as a potential alternative to the popular entropic regularization in computational optimal transport, offering the theoretical advantage of producing sparse couplings through its hinge density structure. Despite recent progress in one-dimensional settings and general upper bounds, fundamental questions about the localization rate of QOT optimizers around the Monge coupling have remained open. In this work, we establish a general lower bound showing that the support of the QOT optimizer cannot concentrate around the Monge graph faster than order $\varepsilon^{\frac{1}{d+2}}$ in the directed Hausdorff distance, matching the conjectured optimal exponent under standard regularity assumptions in Wiesel & Xu (2025). We also show that the QOT value gap controls the mean-squared deviation $\mathbb{E}_{\pi_\varepsilon}\|y - T(x)\|^2$ by the scale of $\varepsilon^{\frac{2}{d+2}}$. As a corollary, in the affine Brenier regime, which includes Gaussian-to-Gaussian transport, we derive a sharp pointwise tube bound of order $\varepsilon^{\frac{1}{d+2}}$ by reducing the problem to self-transport and applying recent self-transport sparsity results. Finally, we validate our theoretical bound with a synthetic experiment in high-dimensional settings.

## 1. Introduction

The optimal transport (OT) problem concerns the task of moving probability measures from a source to a target location (Monge, 1781). It has seen multiple applications in machine learning in recent years, including distribution alignment for (unsupervised) domain adaptation (Courty et al., 2016), training generative models with Wasser-stein/Sinkhorn OT losses (Arjovsky et al., 2017; Genevay et al., 2018), and using Wasserstein-based losses in structured prediction (including multi-label learning) (Frogner et al., 2015). OT also appears in modern generative modeling: flow matching can be instantiated with OT displacement interpolation (Wasserstein geodesics) as the probability path (Lipman et al., 2023), and related ODE-based transport methods (e.g., rectified-flow variants) are explicitly OT-motivated (Liu, 2022). In computational biology, OT-based models are used for single-cell data integration and for learning perturbation responses (Schiebinger et al., 2019; Bunne et al., 2023; 2024). In NLP (Natural Language Processing), the Word Mover's Distance casts document comparison as an OT problem over word embeddings (Kusner et al., 2015).

Formally, let $\mu, \nu \in \mathcal{P}(\mathbb{R}^d)$ be Borel probability measures and let $c : \mathbb{R}^d \times \mathbb{R}^d \to \mathbb{R}$ be a measurable cost. In Monge's original formulation, one seeks a measurable map $T : \mathbb{R}^d \to \mathbb{R}^d$ pushing $\mu$ to $\nu$ (i.e. $T_\#\mu = \nu$) and minimizing

$$\mathrm{OT}(\mu, \nu) \coloneqq \inf_{T : T_\#\mu = \nu} \int c(x, T(x)) \mu(dx),$$

where $T_\#\mu(B) \coloneqq \mu(T^{-1}(B))$ for all Borel sets $B \subseteq \mathbb{R}^d$. The Kantorovich optimal transport formulation (Kantorovich, 1942) relaxes the mapping constraint by couplings and yields the convex problem

$$\mathrm{OT}(\mu, \nu) \coloneqq \inf_{\pi \in \Pi(\mu, \nu)} \int c(x, y) \pi(dx, dy),$$

where $\Pi(\mu, \nu)$ is the set of couplings with marginal $\mu, \nu$:

$$\Pi(\mu, \nu) \coloneqq \{\pi \in \mathcal{P}(\mathbb{R}^d \times \mathbb{R}^d) : \pi(\cdot \times \mathbb{R}^d) = \mu,$$
$$\pi(\mathbb{R}^d \times \cdot) = \nu\}.$$

Throughout this paper, we focus on the quadratic cost $c(x, y) = \frac{1}{2}\|x - y\|^2$, where $\|\cdot\|$ denotes the Euclidean norm. In this case, if $\mu \ll \mathrm{Leb}$, and $\mu, \nu$ have finite second moments, Brenier's theorem (Brenier, 1991) asserts that the Monge and Kantorovich problems coincide and admit a unique optimizer, which means, there exists a convex function $\varphi : \mathbb{R}^d \to \mathbb{R}$ such that the map $T \coloneqq \nabla\varphi$ satisfies $T_\#\mu = \nu$ and the optimal plan is $\pi_\star = (\mathrm{Id}, T)_\#\mu$. We refer the readers to Villani et al. (2008) for background and further details.

[1]College of Engineering & Computer Science, VinUniversity [2]Center for AI Research, VinUniversity, Hanoi, Vietnam. Correspondence to: Long Nguyen-Chi <lc.stilldomath@gmail.com>, Binh T. Nguyen <binh.nt2@vinuni.edu.vn>.

*Proceedings of the 43rd International Conference on Machine Learning*, Seoul, South Korea. PMLR 306, 2026. Copyright 2026 by the author(s).

**Regularization and sparsity.** Regularization is ubiquitous in computational OT and in machine learning applications by improving the scalability of the OT problem (Peyré & Cuturi, 2019). A classical choice is entropic regularization (Cuturi, 2013), which leads to Sinkhorn-type algorithms but produces dense couplings: the optimizer $\pi_\varepsilon^{\text{ent}}$ has a strictly positive density w.r.t $P = \mu \otimes \nu$. Quadratically regularized optimal transport (QOT) replaces the entropic penalty by a squared $L^2(\mu \otimes \nu)$ penalty (equivalently a $\chi^2$ divergence up to an additive constant), and has been advocated as a sparsity-promoting alternative in both discrete and continuous settings (Blondel et al., 2018; Seguy et al., 2018; Essid & Solomon, 2018; Lorenz et al., 2021). A key structural difference from entropic OT is that the QOT optimizer admits a hinge (truncation) density formula (cf. (2.2)), in contrast to the strictly positive Gibbs density of entropic OT. This hinge structure produces exact zeros whenever the dual slack is negative, and therefore directly exposes sparsity through strict inequalities in the dual slack (Lorenz et al., 2021; Nutz, 2025).

More precisely, fix $\varepsilon > 0$ and set the reference measure $P := \mu \otimes \nu$. For $\pi \ll P$, we study the $\chi^2/L^2$ (quadratic) regularization

$$\text{QOT}_\varepsilon(\mu, \nu) :=$$
$$\inf_{\pi \in \Pi(\mu,\nu)} \left\{ \int c(x,y) d\pi(x,y) + \frac{\varepsilon}{2} \left\| \frac{d\pi}{dP} \right\|_{L^2(P)}^2 \right\}, \quad (1.1)$$

which differs from the $\chi^2$-divergence $D_{\chi^2}(\pi \| P) = \int (\frac{d\pi}{dP} - 1)^2 dP$ only by an additive constant in the objective. Here, $\varepsilon > 0$ is the regularization parameter and $d\pi/dP$ denotes the Radon-Nikodym derivative.

**Localization around the Monge coupling.** We denote by $\pi_\varepsilon$ the unique optimizer of the QOT problem (1.1). A natural question is: as $\varepsilon \downarrow 0$, how fast does the optimizer $\pi_\varepsilon$ concentrate around the Monge graph $\text{gr}\,T$? This question quantifies the intrinsic sparsity-accuracy tradeoff of QOT: the hinge density promotes sparse support, but a small $\varepsilon$ is needed for $\pi_\varepsilon$ to approximate the Monge coupling $\pi_\star$. Controlling the tube thickness around $\text{gr}\,T$ is also practically relevant when $\pi_\varepsilon$ is used as a sparse correspondence/graph (or via barycentric projection to estimate transport maps), rather than only through its scalar cost value (Matsumoto et al., 2022; Pooladian & Niles-Weed, 2021). Recent statistical analyses of sparsity-promoting regularized OT further suggest that quantitative control of the geometry of $\text{spt}\,\pi_\varepsilon$ is a key input for obtaining parametric sample complexity for sparse regularizers, including quadratic regularization (González-Sanz et al., 2025a;b).

We quantify this by the directed Hausdorff distance and vertical bias

$$r := \text{dist}(\text{spt}\,\pi_\varepsilon; \text{gr}\,T), b := \sup_{(x,y) \in \text{spt}\,\pi_\varepsilon} \|y - T(x)\|.$$

When $T$ is $L$-Lipschitz, $r \le b \le \sqrt{1 + L^2}r$ (Lemma 3.1). Recent work has begun to quantify how $\text{spt}\,\pi_\varepsilon$ concentrates around the Monge coupling as $\varepsilon \downarrow 0$, obtaining sharp rates in one dimension and the first general-dimensional upper bounds under geometric and spread assumptions (González-Sanz & Nutz, 2024; Wiesel & Xu, 2025; Gvalani & Koch, 2026). However, outside the self-transport case $\mu = \nu$, the available general-dimensional upper bounds do not match the conjectured $\varepsilon^{1/(d+2)}$ scale under standard assumptions. Moreover, prior to this work, a general lower bound ruling out faster-than-$\varepsilon^{1/(d+2)}$ concentration was not available. Our contribution can be summarized as follows.

**Contributions.** We provide the following results for the QOT problem with the quadratic cost $c(x,y) := \frac{1}{2}\|x - y\|^2$.

1. **A general lower bound with the conjectured exponent.** Under standard regularity assumptions on $\mu, \nu$ and the Lipschitzness of the Monge map $T$, we prove that $\text{spt}\,\pi_\varepsilon$ cannot concentrate around $\text{gr}\,T$ faster than order $\varepsilon^{1/(d+2)}$ in directed Hausdorff distance (Theorem 3.3).

2. **Mean-squared and high-probability bias from the QOT value gap.** We show that the QOT value gap $\Delta_\varepsilon$ upper bounds the mean-squared deviation from the Monge map under $\pi_\varepsilon$ (Lemma 3.5). Combining this with sharp value-gap rates from Eckstein & Nutz (2024) yields the $\varepsilon^{2/(d+2)}$ mean-square scale and a simple tail bound for $\|Y - T(X)\|$ when $(X, Y) \sim \pi_\varepsilon$ (Theorem 3.6).

3. **Affine regime: sharp pointwise upper bound.** When $T$ is affine ($T(x) = Ax + b$ with $A \succ 0$, including Gaussian-to-Gaussian transport), we obtain the sharp pointwise bias upper bound $\sup_{(x,y) \in \text{spt}\,\pi_\varepsilon} \|y - T(x)\| \lesssim \sqrt{\lambda_{\max}(A)}\varepsilon^{1/(d+2)}$ (Theorem 3.7).

**Organization.** Section 2 introduces the background for understanding the whole paper, Section 3 gives the three results above, Section 4 shows the important results in QOT, Section 5 provides synthetic experiments validating the $\varepsilon^{1/(d+2)}$ scaling in the affine Brenier regime, and Section 6 summarizes the whole paper and provides some suggestions for the future works.

## 2. Preliminaries

### 2.1. Notations

For a map $F : \mathbb{R}^d \to \mathbb{R}^d$, we denote its graph by

$$\text{gr}\,F := \{(x, F(x)) : x \in \mathbb{R}^d\}.$$

For a point $m$ in a metric space $(M, d)$ and a set $B \subseteq M$, define

$$\text{dist}(m, B) := \inf_{b \in B} d(m, b).$$

For sets $A, B \subseteq M$, define the directed Hausdorff distance

$$\text{dist}(A; B) := \sup_{a \in A} \text{dist}(a, B) = \sup_{a \in A} \inf_{b \in B} d(a, b).$$

In our setting, we always take $M = \mathbb{R}^d \times \mathbb{R}^d$ with the Euclidean distance. In particular, for a coupling $\pi$ we have:

$$\text{dist}(\text{spt}\,\pi; \text{gr}\,T) = \sup_{(x,y) \in \text{spt}\,\pi} \inf_{x'} \big\| (x, y) - (x', T(x')) \big\|.$$

Thus $\text{dist}(\text{spt}\,\pi; \text{gr}\,T)$ measures how far the support of $\pi$ can deviate from the graph of the Monge map. For a function $f : \mathbb{R}^d \to \mathbb{R} \cup \{+\infty\}$, we denote by

$$f^*(y) := \sup_{x \in \mathbb{R}^d} \big[ \langle x, y \rangle - f(x) \big]$$

its convex conjugate.

Let $B(y, r)$ denote the (closed) Euclidean ball in $\mathbb{R}^d$ centered at $y$ with radius $r > 0$. Let $\omega_d = \pi^{d/2} / \Gamma(\frac{d}{2} + 1)$ be the unit-ball $B(0, 1)$ volume.

## 2.2. Dual formulation of quadratically regularized optimal transport

Seguy et al. (2018) derived the Lagrangian dual of (1.1), and Lorenz et al. (2021) proved strong duality and the existence of dual optimizers. The dual problem associated with (1.1) reads

$$\sup_{f \in L^1(\mu), g \in L^1(\nu)} \bigg\{ \int f(x) d\mu(x) + \int g(y) d\nu(y)$$
$$- \frac{1}{2\varepsilon} \int \big[ f(x) + g(y) - c(x,y) \big]_+^2 dP(x,y) \bigg\}, \quad (2.1)$$

where $[\cdot]_+ := \max\{\cdot, 0\}$, and $(f_\varepsilon, g_\varepsilon)$ denotes any maximizer of (2.1); we call them optimal dual potentials. It is well known from Lorenz et al. (2021) that for some continuous $f_\varepsilon, g_\varepsilon$, the unique optimizer $\pi_\varepsilon$ of (1.1) is absolutely continuous w.r.t $P$, with density

$$\frac{d\pi_\varepsilon}{dP}(x, y) = \frac{1}{\varepsilon} \big[ f_\varepsilon(x) + g_\varepsilon(y) - c(x,y) \big]_+, \quad (2.2)$$

The support of $\pi_\varepsilon$ is:

$$\text{spt}\,\pi_\varepsilon = \overline{\big\{ (x, y) \in \text{spt}\,P : f_\varepsilon(x) + g_\varepsilon(y) > c(x,y) \big\}}. \quad (2.3)$$

Up to constant shifts, they are uniquely defined by

$$\int \big[ f_\varepsilon(x) + g_\varepsilon(y) - c(x,y) \big]_+ \mu(dx) = \varepsilon \ \forall y \in \text{spt}\,\nu, \quad (2.4)$$

$$\int \big[ f_\varepsilon(x) + g_\varepsilon(y) - c(x,y) \big]_+ \nu(dy) = \varepsilon \ \forall x \in \text{spt}\,\mu. \quad (2.5)$$

A key structural feature is that the optimal density is a truncated affine function of the dual slack, cf. (2.2); this directly exposes sparsity through the strict inequality in (2.3).

# 3. Results

Our first main result gives a lower bound on $r$ (hence also on $b$) with exponent $1/(d+2)$, which coincides with the QOT bias conjecture rate. Our second result shows that the value gap $\Delta_\varepsilon := \text{QOT}_\varepsilon(\mu, \nu) - \text{OT}(\mu, \nu)$ controls the mean-squared bias $\int \|y - T(x)\|^2 d\pi_\varepsilon$ and yields an $\varepsilon^{2/(d+2)}$ scale. Finally, in the affine Brenier regime $T(x) = Ax + a$, we reduce $\text{QOT}(\mu, \nu)$ exactly to a self-transport QOT problem and obtain the sharp pointwise upper bound $b \lesssim \sqrt{\lambda_{\max}(A)} \varepsilon^{1/(d+2)}$.

We begin by specifying the standing assumptions on $(\mu, \nu)$ and the Monge map $T$. Note that we work with compactly supported probability measures $\mu$ and $\nu$ on $\mathbb{R}^d$.

**Assumption 1.** There exists a bounded connected open Lipschitz domain $\Omega \subset \mathbb{R}^d$ with $\Omega \subset B(0, 1)$ such that $\mu(dx) = \rho_\mu(x) \mathbf{1}_\Omega(x) dx$, and there are constants $0 < \underline{\lambda}_\mu \leq \overline{\lambda}_\mu < \infty$ such that

$$\underline{\lambda}_\mu \leq \rho_\mu(x) \leq \overline{\lambda}_\mu \text{ for Lebesgue-a.e. } x \in \Omega.$$

This leads to $\text{spt}\,\mu = \overline{\Omega}$.

**Assumption 2.** The measure $\nu \in \mathcal{P}(\mathbb{R}^d)$ satisfies $\nu \ll \text{Leb}$, $\text{spt}\,\nu \subseteq B(0, 1)$ and there exist constants $0 < \overline{\lambda}_\nu < \infty$ such that $d\nu/dy \leq \overline{\lambda}_\nu$ on $\text{spt}\,\nu$.

The normalization $\text{spt}\,\mu, \text{spt}\,\nu \subseteq B(0, 1)$ is convenient and can always be achieved by rescaling. These assumptions ensure, in particular, that $\mu$ and $\nu$ are compactly supported, since supports of Borel probability measures are closed subsets of $\mathbb{R}^d$.

Besides Assumptions 1-2, we use the following standard assumption:

**Assumption 3.** The Monge map $T = \nabla \varphi$ is $L$-Lipschitz for some $L > 0$.

**Remark on assumptions.** Assumption 1-2 (bounded Lipschitz domain and density bounded above/below) are standard "no-holes, no-vanishing-density" conditions. This is the typical regime in which the $\varepsilon$-spread scales as $\delta(\varepsilon) \asymp \varepsilon^{1/(d+1)}$, matching the canonical benchmark in previous work on QOT sparsity. Here, we define the $\varepsilon$-spread of $\mu$ by

$$\delta(\varepsilon) := \inf\{ r > 0 : r\rho(r) > \varepsilon \},$$
$$\rho(r) := \inf_{x \in \text{spt}\,\mu} \mu\big( B(x, r) \big).$$

where $B(x, r)$ denotes the closed Euclidean ball of radius $r$ centered at $x$. Intuitively, $\delta(\varepsilon)$ captures how uniformly

the probability mass of $\mu$ is spread over its support: if every ball of radius $r$ inside the support carries at least some fixed amount of mass, then $\rho(r)$ is not too small, and therefore $\delta(\varepsilon)$ is also small.

This type of assumption is standard in statistical guarantees for regularized OT maps; see, e.g., Pooladian & Niles-Weed (2021). The "density bounded from 0 and infinity" assumption is also used by González-Sanz & Nutz (2024) to show the sharp rate of the distance to the optimizer in 1D. Assumption 3 mirrors the regularity used in the bias upper bound of Wiesel & Xu (2025) and in statistical analyses of OT map (Hütter & Rigollet, 2021; Muzellec et al., 2024; Balakrishnan & Manole, 2025).

We now derive a nontrivial lower bound that does not suffer from a global subtraction term and matches the optimal exponent in $\varepsilon$.

### 3.1. A general lower bound to the Monge map

We first state the following elementary bound on distance to a Lipschitz graph.

**Lemma 3.1** (Distance to a Lipschitz graph). *Let $T : \mathbb{R}^d \to \mathbb{R}^d$ be $L$-Lipschitz. Then for any $(x, y) \in \mathbb{R}^d \times \mathbb{R}^d$,*

$$\mathrm{dist}\big((x, y); \mathrm{gr}\, T\big) \geq \frac{\|y - T(x)\|}{(1 + L^2)^{1/2}}.$$

**Remark on distance vs bias.** If $T$ is $L$-Lipschitz, then for all $(x, y)$,

$$\mathrm{dist}\big((x, y); \mathrm{gr}\, T\big) \leq \|y - T(x)\|$$
$$\leq \sqrt{1 + L^2}\, \mathrm{dist}\big((x, y); \mathrm{gr}\, T\big).$$

Hence the directed distance $\mathrm{dist}(\mathrm{spt}\, \pi_\varepsilon; \mathrm{gr}\, T)$ and the vertical bias $\sup_{(x,y)\in\mathrm{spt}\,\pi_\varepsilon} \|y - T(x)\|$ are equivalent up to the factor $\sqrt{1 + L^2}$.

**Lemma 3.2** (Fiberwise $L^2$ lower bound by marginal constraints). *Let $\pi_\varepsilon \ll P$ with density $h = \frac{d\pi_\varepsilon}{dP}$ and let $r := \mathrm{dist}(\mathrm{spt}\,\pi_\varepsilon; \mathrm{gr}\, T)$. For every $y$, define $X_y := \{x \in \mathrm{spt}\,\mu : h(x, y) > 0\}$. Then:*

*(i)* $\int h(x, y)^2 \mu(dx) \geq \frac{1}{\mu(X_y)}$

*(ii) If $T$ is $L$-Lipschitz, then*

$$X_y \subseteq T^{-1}\big(B(y, (1 + L^2)^{1/2}r)\big) \text{ up to a } \mu\text{-null set,}$$

*hence $\mu(X_y) \leq \nu\big(B(y, (1 + L^2)^{1/2}r)\big)$ and, under Assumption 2,*

$$\mu(X_y) \leq \overline{\lambda}_\nu \mathrm{Leb}\big(B(0, (1 + L^2)^{1/2}r)\big)$$
$$= \overline{\lambda}_\nu (1 + L^2)^{d/2} \omega_d r^d.$$

*Consequently,*

$$\|h\|_{L^2(P)}^2 = \int\int h^2 d\mu d\nu \geq \int \frac{1}{\mu(X_y)}\nu(dy)$$
$$\geq \frac{1}{\overline{\lambda}_\nu (1 + L^2)^{d/2}\omega_d} r^{-d}. \tag{3.1}$$

Recall the value gap

$$\Delta_\varepsilon = \mathrm{QOT}_\varepsilon(\mu, \nu) - \mathrm{OT}(\mu, \nu) \geq 0.$$

For the $\chi^2$ (i.e., $L^2$) regularization, because $\mathrm{spt}\,\mu, \mathrm{spt}\,\nu \subseteq B(0, 1)$, we have $\mu, \nu \in \mathcal{P}_{2+\delta}(\mathbb{R}^d)$ for every $\delta > 0$. So, we can use Eckstein & Nutz, 2024, Corollary 3.1, Proposition 4.2, which gave convergence rates in the quadratic-cost case:

$$C_{\mathrm{val}}^{-1}\varepsilon^{\frac{2}{2+d}} \leq \Delta_\varepsilon \leq C_{\mathrm{val}}\varepsilon^{\frac{2}{2+d}} \tag{3.2}$$

for $\varepsilon \in (0, 1]$.

Lemma 3.2 gives the lower bound for $\|h\|_{L^2(P)}^2$. Since the quadratic penalty contributes $(\varepsilon/2)\|h\|_{L^2(P)}^2$ to $\mathrm{QOT}_\varepsilon$, we can get the lower bound of $r$, proved in Theorem 3.3.

**Theorem 3.3** (General lower bound with optimal exponent). *Let $\pi_\varepsilon$ be the QOT optimizer and let Assumptions 1,2, and 3 hold. Set*

$$C_0 := \overline{\lambda}_\nu (1 + L^2)^{d/2}\omega_d$$

*and*

$$C_{\mathrm{lin}} := \int cdP - \mathrm{OT}(\mu, \nu) + \frac{1}{2}.$$

*Then there exist constants*

$$c_{\mathrm{sm}} = \left(\frac{1}{2C_0 C_{\mathrm{val}}}\right)^{1/d}, c_{\mathrm{lg}} = \left(\frac{1}{2C_0 C_{\mathrm{lin}}}\right)^{1/d} > 0$$

*(depending only on $d, L, \overline{\lambda}_\nu$ and $C_{\mathrm{val}}$) such that for all $\varepsilon > 0$,*

$$\mathrm{dist}\big(\mathrm{spt}\,\pi_\varepsilon; \mathrm{gr}\, T\big) \geq \begin{cases} c_{\mathrm{sm}}\varepsilon^{\frac{1}{2+d}}, & \varepsilon \in (0, 1], \\ c_{\mathrm{lg}}, & \varepsilon \geq 1. \end{cases} \tag{3.3}$$

*In particular, $\mathrm{dist}(\mathrm{spt}\,\pi_\varepsilon; \mathrm{gr}\, T) \geq \min\{c_{\mathrm{sm}}\varepsilon^{\frac{1}{2+d}}, c_{\mathrm{lg}}\}$ for all $\varepsilon > 0$.*

**Proof sketch.** When $\varepsilon \in (0, 1]$, we combine this with the sharp upper bound on the value gap to obtain the first result. When $\varepsilon \geq 1$, we use the competitor $\pi = P = \mu \otimes \nu$ in (1.1), giving $\mathrm{QOT}_\varepsilon(\mu, \nu) \leq \int cdP + \varepsilon/2$.

We focus on the small $\varepsilon$ regime, so we get this corollary:

**Corollary 3.4.** *Under the assumptions of Theorem 3.3, for every $\varepsilon \in (0, 1]$,*

$$\mathrm{dist}(\mathrm{spt}\,\pi_\varepsilon; \mathrm{gr}\, T) \geq c_{\mathrm{sm}}\varepsilon^{\frac{1}{d+2}}$$

*and hence*

$$\sup_{(x,y)\in\mathrm{spt}\,\pi_\varepsilon} \|y - T(x)\| \geq c_{\mathrm{sm}}\varepsilon^{\frac{1}{d+2}}.$$

## 3.2. A value-gap bound on the mean-squared bias

This subsection provides a simple inequality that converts the QOT value gap $\Delta_\varepsilon$ convergence into map-level control.

**Lemma 3.5** (Value gap controls mean-squared bias). *Assume $\varphi$ is differentiable and Assumption 3 holds. Let $\pi_\varepsilon$ be the optimizer of* (1.1). *Then*

$$\int \|y - \nabla\varphi(x)\|^2 \pi_\varepsilon(dx, dy) \leq 2L\Delta_\varepsilon \qquad (3.4)$$

*In fact, the stronger inequality holds:*

$$\int \|y - \nabla\varphi(x)\|^2 \pi(dx, dy) \leq 2L\Big(\int c\, d\pi - \mathrm{OT}(\mu, \nu)\Big)$$
$$\forall \pi \in \Pi(\mu, \nu),$$

*and* (3.4) *follows since $\int c\, d\pi_\varepsilon - \mathrm{OT}(\mu, \nu) \leq \Delta_\varepsilon$.*

**Proof sketch.** The cost gap $\int c\, d\pi - \mathrm{OT}(\mu, \nu)$ equals $\int D_\varphi d\pi$, where $D_\varphi(x, y) = \varphi(x) + \varphi^*(y) - \langle x, y \rangle$ is the Fenchel-Young slack. If $\nabla\varphi$ is $L$-Lipschitz, smoothness implies the pointwise bound $D_\varphi(x, y) \geq \|y - \nabla\varphi(x)\|^2/(2L)$; integrating gives the claim.

We also prove that this is the sharp bound in $\varepsilon$ for the mean-squared bias through Lemma A.1. Lemma 3.5 yields the sharp $\varepsilon^{2/(d+2)}$ mean-square scale and, by a Markov argument, a corresponding high-probability bound under $\pi_\varepsilon$, which helps us control how most of the weight of $\pi_\varepsilon$ spreads, shown in Theorem 3.6 below.

**Theorem 3.6** (Mean-squared and high-probability bias from the value gap). *Assume Assumptions 1, 2, and 3 hold. For every $t > 0$ and $\varepsilon \in (0, 1]$,*

$$\pi_\varepsilon\Big(\{(x, y) : \|y - T(x)\| \geq t\}\Big) \leq \frac{2LC_{\mathrm{val}}}{t^2} \varepsilon^{\frac{2}{d+2}}. \quad (3.5)$$

*Equivalently, for every $\eta \in (0, 1)$, choosing $t = \sqrt{2LC_{\mathrm{val}}}\eta^{-1/2}\varepsilon^{1/(d+2)}$ yields*

$$\pi_\varepsilon\Big(\{(x, y) : \|y - T(x)\| \leq \sqrt{2LC_{\mathrm{val}}}\eta^{-1/2}\varepsilon^{\frac{1}{d+2}}\}\Big) \geq 1 - \eta.$$

The bound (3.5) is a tail bound under the measure $\pi_\varepsilon$ and does not directly control $\sup_{(x,y)\in\mathrm{spt}\,\pi_\varepsilon} \|y - T(x)\|$.

## 3.3. Affine Monge maps (Gaussian case): reduction to self-transport and sharp pointwise bias

This subsection gives the $\varepsilon^{1/(d+2)}$ pointwise bias in the regime where the Monge map is known and affine, which includes some truncated Gaussian-Gaussian case. The main point is that in the affine Brenier regime, the quadratically regularized problem reduces exactly to a self-transport problem, and we can then invoke the sharp self-transport tube thickness of Wiesel & Xu (2025). In this subsection, we assume that:

**Assumption 4.** The Monge map is affine:

$$T(x) = Ax + a, \, A \in S_{++}^d, a \in \mathbb{R}^d.$$

This is a stronger version of Assumption 3. Assumption 4 holds, for instance, when $\mu, \nu$ are nondegenerate Gaussians (then $T$ is affine with $A \in S_{++}^d$).

We have the following theorem.

**Theorem 3.7** (Affine case). *Assume Assumption 1 and Assumption 4. Let $\mu_A := (A^{1/2})_\# \mu$ and $\Omega_A := A^{1/2}\Omega$. Fix any $r_A > 0$ (e.g. any radius below the Lipschitz localization scale of $\Omega_A$) and define*

$$\kappa_A := \inf_{u \in \mathrm{spt}\,\mu_A} \inf_{0 < r \leq r_A} \frac{\mathrm{Leb}(\Omega_A \cap B(u, r))}{\omega_d r^d} \in (0, 1],$$

$$\underline{\lambda}_{\mu_A} := \frac{\underline{\lambda}_\mu}{\sqrt{\det A}}.$$

*Set*

$$\varepsilon_0 := \underline{\lambda}_{\mu_A} \kappa_A \omega_d r_A^{d+2} = \frac{\underline{\lambda}_\mu}{\sqrt{\det A}} \kappa_A \omega_d r_A^{d+2}.$$

*Then for all $\varepsilon \in (0, \varepsilon_0]$,*

$$\sup_{(x,y)\in\mathrm{spt}\,\pi_\varepsilon} \|y - T(x)\| \leq 8\sqrt{\lambda_{\max}(A)} \left(\frac{\varepsilon}{\underline{\lambda}_{\mu_A}\kappa_A\omega_d}\right)^{\frac{1}{d+2}}$$

$$= 8\sqrt{\lambda_{\max}(A)} \left(\frac{\sqrt{\det A}}{\underline{\lambda}_\mu \kappa_A \omega_d}\right)^{\frac{1}{d+2}} \varepsilon^{\frac{1}{d+2}}. \quad (3.6)$$

*Equivalently, using $\omega_d = \pi^{d/2}/\Gamma(\frac{d}{2} + 1)$,*

$$\sup_{(x,y)\in\mathrm{spt}\,\pi_\varepsilon} \|y - T(x)\| \leq 8\sqrt{\lambda_{\max}(A)}$$

$$\cdot \left(\frac{\sqrt{\det A}\,\Gamma(\frac{d}{2} + 1)}{\underline{\lambda}_\mu \kappa_A \pi^{d/2}}\right)^{\frac{1}{d+2}} \varepsilon^{\frac{1}{d+2}}.$$

**Proof sketch.** In the affine Brenier regime, we can "straighten" the Monge graph into the diagonal and reduce $\mathrm{QOT}(\mu, \nu)$ to a self-transport QOT problem, to which sharp tube bounds in Wiesel & Xu, 2025, Theorem 3.1, 3.4 apply.

For nondegenerate untruncated Gaussians on $\mathbb{R}^d$, the Monge map for the quadratic cost is affine. However, if one truncates Gaussians to bounded domains to enforce compact support, which are needed to satisfy Assumption 1-2, the Monge map is typically not affine unless the truncations are chosen compatibly with the affine map.

To make this precise, let $\Omega_0, \Omega_1 \subset \mathbb{R}^d$ be bounded Lipschitz domains with nonempty interior and let $\Sigma_0, \Sigma_1 \in \mathbb{S}_{++}^d$

and $m_0, m_1 \in \mathbb{R}^d$. Define the (renormalized) truncated Gaussians

$$d\mu(x) = \frac{1}{Z_0} \exp\left(-\frac{1}{2}(x-m_0)^\top \Sigma_0^{-1}(x-m_0)\right)$$
$$\cdot \mathbf{1}_{\Omega_0}(x)dx, \tag{3.7}$$

$$d\nu(y) = \frac{1}{Z_1} \exp\left(-\frac{1}{2}(y-m_1)^\top \Sigma_1^{-1}(y-m_1)\right)$$
$$\cdot \mathbf{1}_{\Omega_1}(y)dy, \tag{3.8}$$

where $Z_i > 0$ are normalizing constants.

**Proposition 3.8** (Affine Monge maps between truncated Gaussians: necessary and sufficient conditions). *Consider $\mu, \nu$ as in* (3.7)-(3.8) *and the quadratic cost $c(x,y) = \frac{1}{2}\|x-y\|^2$. Assume that the (unregularized) Monge map transporting $\mu$ to $\nu$ is affine:*

$$T(x) = Ax + a, \quad A = A^\top \succ 0, a \in \mathbb{R}^d. \tag{3.9}$$

*Then necessarily*

$$\Omega_1 = A\Omega_0 + a \text{ (up to Lebesgue-null sets)},$$
$$m_1 = Am_0 + a, \tag{3.10}$$
$$\Sigma_1 = A\Sigma_0 A.$$

*Conversely, if* (3.10) *holds, then $T(x) = Ax + a$ satisfies $T_{\#}\mu = \nu$ and, since $T = \nabla\left(\frac{1}{2}x^\top Ax + a^\top x\right)$ is the gradient of a convex function, it is the Monge map.*

**Proof sketch.** If $T(x) = Ax + a$ transports $\mu$ to $\nu$, then the pushforward density identity forces the indicator sets to match, hence $\Omega_1 = A\Omega_0 + a$. On $\Omega_1$, the Gaussian factors must agree; taking logarithms yields an identity between two quadratic polynomials in $y$, so matching coefficients gives $m_1 = Am_0 + a$ and $\Sigma_1 = A\Sigma_0 A$. Conversely, under these compatibility conditions, $T_{\#}\mu = \nu$ by change of variables and $T = \nabla(\frac{1}{2}x^\top Ax + a^\top x)$ is a Monge map.

Therefore, truncating both Gaussians on the same bounded domain (e.g., a large ball) generally destroys Assumption 4 unless the two truncated laws coincide. To retain Assumption 4 within a compact-support framework, one should truncate $\mu$ on a domain $\Omega_0$ and choose $\nu$ so that $\Omega_1 = T(\Omega_0)$ and $(m_1, \Sigma_1) = (Am_0 + a, A\Sigma_0 A)$, i.e. $\nu = T_{\#}\mu$, so that the Monge map remains exactly $T(x) = Ax + a$.

## 4. Related works

### 4.1. Quadratically regularized OT (QOT)'s structure, potentials, and behavior

Quadratic regularization of OT has been studied both for computational reasons and as a means to obtain sparse couplings. Blondel et al. (2018) is one of the earliest to propose applying strongly convex primal regularization (including squared $\ell^2$ penalties) to promote sparsity. In the

discrete/graph setting, Essid & Solomon (2018) analyzed quadratically regularized OT (minimum-cost flows) and exploited its second-order dual structure to derive Newton-type algorithms, with a detailed description of how the solution structure changes as the regularization vanishes.

On the asymptotic side, Garriz-Molina et al. (2024) connected the infinitesimal behavior of QOT to porous-medium-type self-similar profiles. Finally, Nutz (2025) proved the existence of QOT potentials under square-integrable costs, analyzed non-uniqueness phenomena (linked to sparsity), and provided qualitative results of sparsity in continuous settings.

**Relation to other regularization schemes.** The tube-versus-density mechanism used in Theorem 3.3 is not specific to the quadratic exponent. Using Hölder's inequality instead of the Cauchy-Schwarz step in Lemma 3.2, we get the lower bound $r^{-d(p-1)}$ for the $L^p$ norm of the density. Thus any sharp value-gap upper bound for the corresponding $L^p$-regularized problem can be converted into a support lower bound by the same argument. The Orlicz-space framework of Lorenz & Mahler (2022) provides the natural functional-analytic setting for such power-type penalties.

ENOT is different in nature: it regularizes the dual Kantorovich potentials for neural optimal transport rather than imposing a primal $f$-divergence penalty on couplings with respect to $\mu \otimes \nu$ (Buzun et al., 2024). Therefore, the support statements proved here do not transfer directly to ENOT. Nevertheless, the hinge/slack viewpoint in QOT suggests a useful conceptual bridge: regularizing positive dual infeasibility can control bias while preserving sparsity or near-sparsity, but the exact localization exponent for ENOT requires a separate analysis of its dual-side objective.

A fundamental quantitative object is the value gap $\Delta_\varepsilon = \text{QOT}_\varepsilon(\mu, \nu) - \text{OT}(\mu, \nu)$. Under quantization-type assumptions, Eckstein & Nutz (2024) derived convergence rates for a broad class of regularizers; specializing to $L^2$ regularization yields

$$\Delta_\varepsilon = \Theta\left(\varepsilon^{\frac{2}{d+2}}\right)(\varepsilon \downarrow 0),$$

which is the scaling used throughout our analysis.

Recent work shows that sparsity-promoting regularization need not forfeit parametric statistical rates. González-Sanz et al. (2025a) established parametric sample complexity and CLTs for QOT costs, couplings, and dual potentials. This further motivates quantitative localization bounds such as ours, since geometric control of $\text{spt}\,\pi_\varepsilon$ directly enters both map estimation and statistical complexity arguments.

### 4.2. Known quantitative bounds for QOT

We briefly recall the main quantitative bounds on the support of $\pi_\varepsilon$, which serve as benchmarks for our results (González-

Sanz & Nutz, 2024; Wiesel & Xu, 2025).

**One-dimensional case.** A key feature of quadratically regularized OT is that optimal couplings are typically supported on a thickened graph. To quantify this, we look at vertical fibers over $x$ and their diameters. For a coupling $\pi \in \mathcal{P}(\mathbb{R}^d \times \mathbb{R}^d)$ and $x \in \mathbb{R}^d$, the vertical fiber of $\operatorname{spt} \pi$ over $x$ is

$$\mathcal{Y}_\pi(x) := \{y \in \mathbb{R}^d : (x, y) \in \operatorname{spt} \pi\},$$

with (fiber) thickness

$$\tau_\pi(x) := \operatorname{diam} \mathcal{Y}_\pi(x) = \sup_{y_1, y_2 \in \mathcal{Y}_\pi(x)} \|y_1 - y_2\|.$$

When the coupling is the quadratically regularized optimizer $\pi_\varepsilon$ (defined below), we abbreviate $\mathcal{Y}_\varepsilon(x) := \mathcal{Y}_{\pi_\varepsilon}(x)$ and $\tau_\varepsilon(x) := \tau_{\pi_\varepsilon}(x)$.

When $d = 1$, if $\mu, \nu$ have continuous densities bounded away from 0 and $\infty$, González-Sanz & Nutz (2024) showed the sharp rate

$$\sup_x \tau_\varepsilon(x) = \Theta(\varepsilon^{1/3}), \operatorname{dist}(\operatorname{spt} \pi_\varepsilon; \operatorname{gr} T) = \Theta(\varepsilon^{1/3}),$$

and establish uniform strong convexity bounds for $f_\varepsilon$ for all $\varepsilon < \varepsilon_0$, here $\varepsilon_0$ is a small constant.

**General dimension case.** Wiesel & Xu (2025) proved the first quantitative pointwise bounds on the bias relative to the Monge map. If $\operatorname{spt} \mu$ is star-shaped and $T$ is $L$-Lipschitz (where $\varphi$ is the Brenier potential for $(\mu, \nu)$), the bias relative to the Monge map satisfies

$$\sup_{(x,y) \in \operatorname{spt} \pi_\varepsilon} \|y - T(x)\| \leq C(L+1)^{3/2} \sqrt[4]{\delta\left(\frac{\delta(\varepsilon)}{L+1}\right)}$$

Here, $\operatorname{spt} \mu$ is star-shaped if there exists $x' \in \operatorname{spt} \mu$ such that for all $x \in \operatorname{spt} \mu$, the line segment from $x'$ to $x$ lies in $\operatorname{spt} \mu$. If $\mu$ is comparable to the uniform measure on the unit ball $\mu = \operatorname{Unif}(B(0,1))$ (that is, $\mu$ admits a density bounded away from 0 and $\infty$ on a set with Lipschitz boundary), then $\rho(r) \asymp r^d$ and thus $\delta(\varepsilon) \asymp \varepsilon^{1/(1+d)}$. Inserting this into the bound above yields the rate

$$\sup_{(x,y) \in \operatorname{spt} \pi_\varepsilon} \|y - T(x)\| \lesssim \varepsilon^{\frac{1}{4(d+1)^2}},$$

which is far from the conjectured $\varepsilon^{1/(d+2)}$ scale in general case.

In the self-transport case $\mu = \nu$ where the optimal coupling is $\operatorname{spt} \pi_* = \{(x, x) : x \in \operatorname{spt} \mu\}$, by introducing the improved $\varepsilon$-spread: $\delta_{\mathrm{ST}}(\varepsilon) := \inf\{r > 0 : r\rho(\sqrt{r}) > \varepsilon\}$, one has the sharp (matching) bounds:

$$\sup_{(x,y) \in \operatorname{spt} \pi_\varepsilon} \|y - T(x)\| \asymp \delta_{\mathrm{ST}}(\varepsilon)^{1/2}.$$

Based on that, when $\mu$ is comparable to $\operatorname{Unif}(B(0,1))$, they obtain matching upper/lower bounds:

$$\sup_{(x,y) \in \operatorname{spt} \pi_\varepsilon} \|y - T(x)\| \asymp \varepsilon^{\frac{1}{2+d}},$$

thereby realizing the conjectured exponent $1/(2+d)$ in full generality under standard spread/Lipschitz conditions.

Compared to these results, our Theorem 3.3 provides a general lower bound with the conjectured exponent $1/(d+2)$ under standard regularity, ruling out any faster-than-$\varepsilon^{1/(d+2)}$ concentration around $\operatorname{gr} T$. Moreover, in the affine Brenier regime (including Gaussian-to-Gaussian transport), our reduction in Section 3.3 combined with Theorem 3.7 yields the matching pointwise bound $b \asymp \varepsilon^{1/(d+2)}$.

## 5. Experiments

We empirically test the tube/bias scaling predicted by Theorem 3.7 in the affine Brenier regime, where the Monge map is known and can be evaluated exactly. Specifically, the theorem predicts that for sufficiently small $\varepsilon$,

$$\sup_{(x,y) \in \operatorname{spt} \pi_\varepsilon} \|y - T(x)\| \lesssim \sqrt{\lambda_{\max}(A)} \varepsilon^{\frac{1}{d+2}}, T(x) = Ax + a.$$

**Affine benchmark with known Monge map.** We construct compactly supported pairs $(\mu, \nu)$ such that $\nu = T_{\#}\mu$ with a prescribed affine map $T(x) = Ax + a$ (cf. Proposition 3.8), so that $T$ is exactly the quadratic-cost Monge map. We discretize $\mu, \nu$ by empirical measures with $N = M = 2000$ samples and uniform weights. (Full details of the truncated-Gaussian family, the choice of $A$, and the discretization protocol are provided in Appendix B.2-B.3.)

**Discrete QOT solver.** For each $\varepsilon$ on a log-spaced grid, we solve the discrete QOT problem (B.4) using two deterministic dual/KKT solvers from Lorenz et al. (2021): nonlinear Gauss-Seidel (Algorithm 1) and a (regularized) semismooth Newton method (Algorithm 3), noted in Appendix B.1. We stop when the maximum absolute marginal residual, denoted in (B.3), is below $\operatorname{tol} = \operatorname{initTol} \cdot \varepsilon$ with $\operatorname{initTol} = 10^{-2}$ (i.e., a relative marginal tolerance of $10^{-2}$).

This corresponds to a relative marginal tolerance that is uniform over $\varepsilon$: indeed, by (B.5) we have

$$\sum_{j=1}^M \pi_{ij} = \frac{a_i}{\varepsilon} \sum_{j=1}^M b_j [f_i + g_j - c_{ij}]_+ = a_i \left(1 + \frac{r_i}{\varepsilon}\right),$$

$$\sum_{i=1}^N \pi_{ij} = b_j \left(1 + \frac{s_j}{\varepsilon}\right),$$

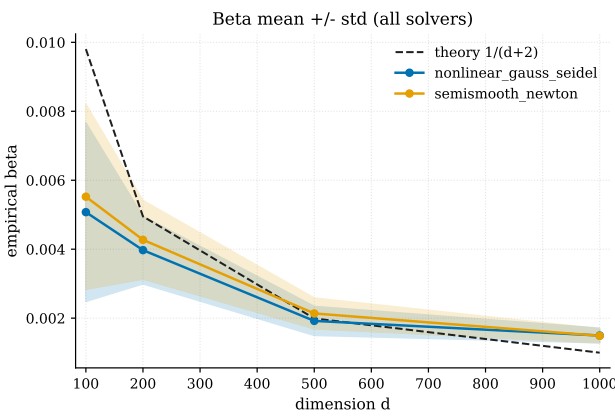
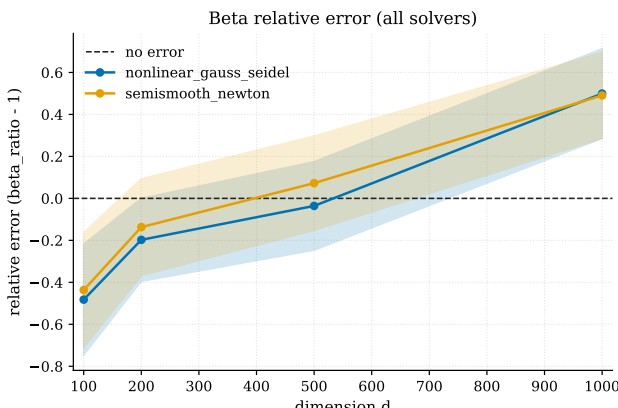

*(a)* Estimated scaling exponent $\widehat{\beta}$ versus the theoretical prediction $1/(d+2)$ from Theorem 3.7.

*(b)* Relative error of the fitted scaling exponent.

*Figure 1.* Affine Brenier scaling diagnostics across dimensions: fitted log-log slope $\widehat{\beta}$ (left) and relative error $(d+2)\widehat{\beta}-1$ (right). Error bars show $\pm 1$ empirical standard deviation over $R=10$ random seeds.

where $(r,s) = F(f,g)$ are the residuals defined in (B.3). Thus $|r_i| \leq \mathrm{initTol}\varepsilon$ and $|s_j| \leq \mathrm{initTol}\varepsilon$ guarantee

$$\left|\sum_j \pi_{ij}-a_i\right| \leq \mathrm{initTol}\cdot a_i \text{ and } \left|\sum_i \pi_{ij}-b_j\right| \leq \mathrm{initTol}\cdot b_j,$$

so the stopping rule enforces a fixed relative error in the marginals across all $\varepsilon$.

Let $\pi_\varepsilon^{N,M}$ denote the unique minimizer of the discrete problem (B.4). To mitigate floating-point noise, we estimate the numerical support using a fixed threshold $\tau = 10^{-12}$ and report the discrete proxy

$$\widehat{\mathrm{bias}}(\varepsilon) := \max_{(i,j):\pi_{\varepsilon,ij}^{N,M}>\tau} \|y_j - T(x_i)\|.$$

**Scaling test.** For each dimension $d \in \{100, 200, 500, 1000\}$ and each random seed, we fit a log-log slope

$$(\widehat{\alpha},\widehat{\beta}) \in \arg\min_{\alpha,\beta} \sum_{k=1}^K \left(\log \widehat{\mathrm{bias}}(\varepsilon_k) - \alpha - \beta\log\varepsilon_k\right)^2,$$

through a grid $(\varepsilon_k)_{k=1}^K$ of $\varepsilon$, and compare $\widehat{\beta}$ to the theoretical exponent $1/(d+2)$. Let $C = (c_{ij})_{1\leq i\leq N,1\leq j\leq M}$ be the discrete cost matrix. We set $K = 10, (\varepsilon_k)_{k=1}^{10} = (10^{-8}, 5\cdot 10^{-8}, 10^{-7}, 5\cdot 10^{-7}, 10^{-6}, 5\cdot 10^{-6}, 10^{-5}, 5\cdot 10^{-5}, 10^{-4}, 5\cdot 10^{-4})$, multiplied by $c_{\mathrm{med}} = \mathrm{median}\{c_{ij} : 1 \leq i \leq N, 1 \leq j \leq M\}$, and we report the mean $\pm$ one empirical standard deviation over $R = 10$ seeds. To directly quantify deviations from the prediction across dimensions, we also report the relative error $\mathrm{RelErr}(d) := \widehat{\beta}/(1/(d+2)) - 1 = (d+2)\widehat{\beta} - 1$ Using $c_{\mathrm{med}}$ provides a robust typical cost scale; a closely related median-normalization is standard in entropic OT practice and implementations (Cuturi, 2013).

**Results.** Figure 1a shows that the fitted exponent $\widehat{\beta}$ decreases with $d$ and that the two deterministic solvers agree within the reported variability. Compared with the theoretical benchmark $1/(d+2)$, the fitted slopes are below the prediction for $d = 100$ and $d = 200$, are closest around $d = 500$ and $d = 1000$. Figure 1b summarizes this behavior via the relative error $(d+2)\widehat{\beta} - 1$, which increases with $d$ and crosses 0 between $d = 500$ and $d = 1000$. We attribute these deviations to finite-sample and non-asymptotic effects: for smaller $d$, the accessible $\varepsilon$ range may not yet be in the asymptotic regime of Theorem 3.7, while for very large $d$ the target exponent $1/(d+2)$ becomes extremely small, making the log-log regression sensitive to discretization floors (finite $N, M$), the support threshold $\tau$, and solver tolerances, which can bias slope estimates upward.

# 6. Conclusion

In this paper, we quantified how the QOT optimizer $\pi_\varepsilon$ localizes around the Monge coupling for the quadratic cost as $\varepsilon \downarrow 0$. First, under standard regularity of the marginals and a Lipschitz Monge map, we proved an information-theoretic lower bound showing that the directed Hausdorff distance from $\mathrm{spt}\,\pi_\varepsilon$ to the Monge graph $\mathrm{gr}\,T$ cannot shrink faster than order $\varepsilon^{1/(d+2)}$. Second, we linked geometry to optimality by showing that the value gap $\Delta_\varepsilon = \mathrm{QOT}_\varepsilon(\mu,\nu) - \mathrm{OT}(\mu,\nu)$ controls the mean-squared deviation from the Monge map under $\pi_\varepsilon$, and we combined this inequality with sharp value-gap rates in Eckstein & Nutz (2024) to obtain the $\varepsilon^{2/(d+2)}$ scale and corresponding high-probability bounds. Third, in the affine Brenier regime (including Gaussian-to-Gaussian transport), we gave an exact reduction to a self-transport QOT problem and obtained a matching pointwise tube bound of order

$\sqrt{\lambda_{\max}(A)}\varepsilon^{1/(d+2)}$ via sharp self-transport sparsity estimates. We also outlined synthetic experiments designed to directly probe this predicted scaling in the affine setting.

**Limitations.** The matching pointwise upper bound proved here is restricted to the affine Brenier regime, so closing the gap between existing general upper bounds and the $\varepsilon^{1/(d+2)}$ barrier in the fully non-affine case remains open.

**Open questions.** A central theoretical direction is to close the remaining gap between general upper bounds and the $\varepsilon^{1/(d+2)}$ lower bound in the fully non-affine setting, ideally under minimal assumptions on $(\mu, \nu)$ and on the regularity of the Brenier potential. From a modeling and computational perspective, extending the analysis to non-compact marginals (e.g., untruncated Gaussians, which can be achieved by extending Wiesel's proof in Wiesel & Xu (2025) to the unbounded support case) and quantifying the effects of truncation and discretization on sparsity and bias would strengthen the connection to practice. However, this is a challenging task because the distributions are no longer compactly supported. We may start with Nutz (2025), where Polish probability spaces are considered.

**Acknowledgments.** The authors would like to thank anonymous reviewers and the Area Chair for their useful feedback to improve our work. This work is supported by grant VUNI.2526.CAIR.04 and Seed Fund 10000225 CECS.

## Impact Statement

This work is theoretical and studies the localization and sparsity properties of quadratically regularized optimal transport. Potential positive impacts include a clearer understanding of sparsity-accuracy tradeoffs in regularized OT objectives that are widely used in machine learning pipelines (e.g., map estimation and correspondence construction). We do not anticipate direct negative societal impacts from the theoretical results themselves; however, as with any methodological advance in machine learning, downstream applications may have dual-use concerns depending on the domain of deployment.

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

## A. Additional proofs

### A.1. Proof of Section 3.1

*Proof of Lemma 3.1.* Fix $(x, y) \in \mathbb{R}^d \times \mathbb{R}^d$. By definition, $\mathrm{dist}\big((x, y); \mathrm{gr}\, T\big) = \inf_{x' \in \mathbb{R}^d} \big\|(x, y) - (x', T(x'))\big\|$. For any $x' \in \mathbb{R}^d$, write

$$\big\|(x, y) - (x', T(x'))\big\|^2 = \|x - x'\|^2 + \|y - T(x')\|^2.$$

Let $a := \|y - T(x)\|$ and $t := \|x - x'\| \geq 0$. By the reverse triangle inequality and Lipschitzness of $T$,

$$\|y - T(x')\| \geq \big|\|y - T(x)\| - \|T(x) - T(x')\|\big| \geq \max\{0, a - Lt\}.$$

Therefore,

$$\big\|(x, y) - (x', T(x'))\big\|^2 \geq t^2 + \big(\max\{0, a - Lt\}\big)^2.$$

If $a \leq Lt$, then $t^2 \geq a^2/L^2 \geq a^2/(1 + L^2)$. If $a > Lt$, then $t^2 + (a - Lt)^2 = (1 + L^2)t^2 - 2Lat + a^2$, whose minimum over $t \geq 0$ is attained at $t^\star = La/(1 + L^2)$ and equals $a^2/(1 + L^2)$. In both cases,

$$\big\|(x, y) - (x', T(x'))\big\|^2 \geq \frac{a^2}{1 + L^2}.$$

Taking the infimum over $x'$ and then the square roots yields the claim. □

*Proof of Lemma 3.2.* For every $y$, we see that the marginal constraint $\int h(x, y)\mu(dx) = 1$ holds. By Cauchy-Schwarz,

$$1 = \int_{X_y} h\, d\mu \leq \left(\int_{X_y} h^2 d\mu\right)^{1/2} \mu(X_y)^{1/2},$$

yielding (i).

For (ii), If $x \in X_y$, then $h(x, y) > 0$. Since $h$ is the continuous hinge representative from (2.2) and $(x, y) \in \mathrm{spt}\, P$, every sufficiently small neighborhood of $(x, y)$ has positive $\pi_\varepsilon$-mass. Hence $(x, y) \in \mathrm{spt}\, \pi_\varepsilon$. With $r' = \mathrm{dist}((x, y); \mathrm{gr}\, T)$, Lemma 3.1 gives $\|y - T(x)\| \leq (1 + L^2)^{1/2} r' \leq (1 + L^2)^{1/2} r$. Hence, for fixed $y$, any $x$ with $h(x, y) > 0$ satisfies $x \in T^{-1}(B(y, (1 + L^2)^{1/2} r))$, proving the inclusion and then the measure bounds by $T_{\#}\mu = \nu$ and Assumption 2. Integrating (i) over $\nu$ and inserting the bound on $\mu(X_y)$ yields (3.1). □

*Proof of Theorem 3.3.* Let $r := \mathrm{dist}(\mathrm{spt}\, \pi_\varepsilon; \mathrm{gr}\, T)$ and $h = \frac{d\pi_\varepsilon}{dP}$. Lemma 3.2 yields $\|h\|_{L^2(P)}^2 \geq \frac{1}{C_0} r^{-d}$. Moreover, $\Delta_\varepsilon \geq \frac{\varepsilon}{2}\|h\|_{L^2(P)}^2$. Hence,

$$\Delta_\varepsilon \geq \frac{\varepsilon}{2C_0} r^{-d} \tag{A.1}$$

Case 1: $\varepsilon \in (0, 1]$. The sharp upper bound $\Delta_\varepsilon \leq C_{\mathrm{val}} \varepsilon^{\frac{2}{2+d}}$ for $\varepsilon \in (0, 1]$ from (3.2) combines with (A.1) to give

$$\frac{\varepsilon}{2C_0} r^{-d} \leq C_{\mathrm{val}} \varepsilon^{\frac{2}{2+d}} \implies r \geq \left(\frac{1}{2C_0 C_{\mathrm{val}}}\right)^{1/d} \varepsilon^{\frac{1}{2+d}} = c_{\mathrm{sm}} \varepsilon^{\frac{1}{2+d}}.$$

Case 2: $\varepsilon \geq 1$. Test (QOT) with the reference $P = \mu \otimes \nu$ (which is admissible) to obtain the upper bound

$$\mathrm{QOT}_\varepsilon(\mu, \nu) \leq \int c\, dP + \frac{\varepsilon}{2},$$

so that

$$\Delta_\varepsilon \leq \left(\int c\, dP - \mathrm{OT}(\mu, \nu)\right) + \frac{\varepsilon}{2} \leq C_{\mathrm{lin}} \varepsilon,$$

where the last inequality uses $\varepsilon \geq 1$. Note that $C_{\mathrm{lin}} < \infty$ because $c$ is bounded on $\mathrm{spt}\, \mu \times \mathrm{spt}\, \nu \subseteq B(0, 1)^2$ and $\mathrm{OT}(\mu, \nu) \geq 0$. Combining this with (A.1) yields

$$\frac{\varepsilon}{2C_0} r^{-d} \leq C_{\mathrm{lin}} \varepsilon \implies r \geq \left(\frac{1}{2C_0 C_{\mathrm{lin}}}\right)^{1/d} = c_{\mathrm{lg}}.$$

This proves (3.3). □

## A.2. Proof of Section 3.2

*Proof of Lemma 3.5.* Define the Fenchel-Young slack

$$D_\varphi(x, y) := \varphi(x) + \varphi^*(y) - \langle x, y \rangle \geq 0.$$

Also set

$$u(x) := \frac{1}{2}\|x\|^2 - \varphi(x), v(y) := \frac{1}{2}\|y\|^2 - \varphi^*(y).$$

Since $c(x, y) = \frac{1}{2}\|x\|^2 + \frac{1}{2}\|y\|^2 - \langle x, y \rangle$, we have

$$c(x, y) - u(x) - v(y) = D_\varphi(x, y).$$

Therefore, for any $\pi \in \Pi(\mu, \nu)$,

$$\int c d\pi - \int u d\mu - \int v d\nu = \int (c - u - v) d\pi = \int D_\varphi d\pi.$$

For the optimal (unregularized) coupling $\pi_\star = (\mathrm{Id}, \nabla\varphi)_\# \mu$, Fenchel-Young is an equality along $(x, \nabla\varphi(x))$, hence $\int u d\mu + \int v d\nu = \int c d\pi_\star = \mathrm{OT}(\mu, \nu)$, and we conclude

$$\int D_\varphi d\pi = \int c d\pi - \mathrm{OT}(\mu, \nu) \ \forall \pi \in \Pi(\mu, \nu). \tag{A.2}$$

Next, we lower bound the slack by the squared bias. Since $\nabla\varphi$ is $L$-Lipschitz, the standard smoothness (descent) inequality gives

$$\varphi(z) \leq \varphi(x) + \langle \nabla\varphi(x), z - x \rangle + \frac{L}{2}\|z - x\|^2.$$

Fix $(x, y)$ and choose $z := x + \frac{1}{L}(y - \nabla\varphi(x))$. Then, by the definition of $\varphi^*$,

$$\varphi^*(y) = \sup_w \{\langle y, w \rangle - \varphi(w)\} \geq \langle y, z \rangle - \varphi(z),$$

and plugging the smoothness bound at $(x, z)$ yields the pointwise inequality

$$D_\varphi(x, y) = \varphi(x) + \varphi^*(y) - \langle x, y \rangle \geq \frac{1}{2L}\|y - \nabla\varphi(x)\|^2. \tag{A.3}$$

This is the standard "smoothness $\Rightarrow$ quadratic lower bound on Fenchel-Young slack". Integrating (A.3) against any $\pi \in \Pi(\mu, \nu)$ and using (A.2) gives

$$\frac{1}{2L}\int \|y - \nabla\varphi(x)\|^2 d\pi \leq \int D_\varphi d\pi = \int c d\pi - \mathrm{OT}(\mu, \nu).$$

Apply this to $\pi = \pi_\varepsilon$ and note $\int c d\pi_\varepsilon - \mathrm{OT}(\mu, \nu) \leq \Delta_\varepsilon$ to obtain (3.4). $\square$

**Lemma A.1** (Optimality of the mean-squared bias scale). *Assume Assumptions 1 and 2 hold. Let*

$$m_\varepsilon := \int \|y - T(x)\|^2 \pi_\varepsilon(dx, dy).$$

*Then, for all $\varepsilon \in (0, 1]$, $m_\varepsilon \geq c_{\mathrm{mse}}\varepsilon^{\frac{2}{d+2}}$, where*

$$c_{\mathrm{mse}} := \frac{1}{2}\left(\frac{1}{96\overline{\lambda}_\nu \omega_d C_{\mathrm{val}}}\right)^{2/d}.$$

*Proof of Lemma A.1.* Let $h = d\pi_\varepsilon/d(\mu \otimes \nu)$ and set $Z(x, y) := \|y - T(x)\|^2$. We first prove that $m_\varepsilon > 0$. The map $T$ is Borel, so $\mathrm{gr}\, T$ is a Borel subset of $\mathbb{R}^d \times \mathbb{R}^d$. If $m_\varepsilon = 0$, then the nonnegative random variable $Z$ vanishes $\pi_\varepsilon$-a.s., and therefore $\pi_\varepsilon(\mathrm{gr}\, T) = 1$. On the other hand, since $\nu \ll \mathrm{Leb}$, every singleton has zero $\nu$-mass. Hence

$$(\mu \otimes \nu)(\mathrm{gr}\, T) = \int \nu(\{T(x)\})\mu(dx) = 0.$$

This contradicts $\pi_\varepsilon \ll \mu \otimes \nu$. Therefore $m_\varepsilon > 0$.

Let $t = \sqrt{2m_\varepsilon}$. By Markov's inequality applied to $Z$,

$$\pi_\varepsilon\big(\{(x,y) : \|y - T(x)\| \le t\}\big) = \pi_\varepsilon(\{Z \le 2m_\varepsilon\}) \ge \frac{1}{2}.$$

For $\nu$-a.e. $y$, define

$$a(y) := \int_{T^{-1}(B(y,t))} h(x,y)\mu(dx).$$

Then $0 \le a(y) \le 1$ for $\nu$-a.e. $y$, and

$$\int a(y)\nu(dy) = \pi_\varepsilon(\{(x,y) : \|y - T(x)\| \le t\}) \ge \frac{1}{2}.$$

Let $D = \{y : a(y) \ge 1/4\}$. Since $a \le 1$,

$$\frac{1}{2} \le \int a(y)\nu(dy) \le \nu(D) + \frac{1}{4}(1 - \nu(D)),$$

and therefore $\nu(D) \ge 1/3$.

For every $y \in D$, Cauchy-Schwarz yields

$$\frac{1}{16} \le \left(\int_{T^{-1}(B(y,t))} h(x,y)\mu(dx)\right)^2 \le \mu(T^{-1}(B(y,t))) \int_{T^{-1}(B(y,t))} h(x,y)^2 \mu(dx).$$

Since $T_{\#}\mu = \nu$ and Assumption 2 gives $\nu(B(y,t)) \le \overline{\lambda}_\nu \omega_d t^d$, we obtain

$$\int h(x,y)^2 \mu(dx) \ge \frac{1}{16\overline{\lambda}_\nu \omega_d t^d} \text{ for every } y \in D.$$

Integrating over $D$ gives

$$\|h\|_{L^2(\mu \otimes \nu)}^2 \ge \frac{1}{48\overline{\lambda}_\nu \omega_d t^d}.$$

Since

$$\Delta_\varepsilon = \int c \, d\pi_\varepsilon - \mathrm{OT}(\mu, \nu) + \frac{\varepsilon}{2}\|h\|_{L^2(\mu \otimes \nu)}^2 \ge \frac{\varepsilon}{2}\|h\|_{L^2(\mu \otimes \nu)}^2,$$

we obtain

$$\Delta_\varepsilon \ge \frac{\varepsilon}{96\overline{\lambda}_\nu \omega_d t^d} = \frac{\varepsilon}{96\overline{\lambda}_\nu \omega_d (2m_\varepsilon)^{d/2}}.$$

Combining this lower bound with $\Delta_\varepsilon \le C_{\mathrm{val}}\varepsilon^{2/(d+2)}$ gives

$$m_\varepsilon^{d/2} \ge \frac{1}{96 \times 2^{d/2}\overline{\lambda}_\nu \omega_d C_{\mathrm{val}}}\varepsilon^{d/(d+2)}.$$

Raising both sides to the power $2/d$ gives the claimed constant $c_{\mathrm{mse}}$. $\qquad\square$

*Proof of Theorem 3.6.* Combining (3.4) with (3.2) yields the sharp mean-square scale

$$\int \|y - T(x)\|^2 \pi_\varepsilon(dx, dy) \le 2LC_{\mathrm{val}}\varepsilon^{\frac{2}{d+2}}.$$

when $\varepsilon \in (0, 1]$. Consequently, for every $t > 0$, Markov's inequality gives a high-probability pointwise bias bound:

$$\pi_\varepsilon\Big(\{(x,y) : \|y - T(x)\| \ge t\}\Big) \le \frac{1}{t^2}\int \|y - T(x)\|^2 d\pi_\varepsilon$$
$$\le \frac{2LC_{\mathrm{val}}\varepsilon^{\frac{2}{d+2}}}{t^2}.$$

$\qquad\square$

## A.3. Proof of Section 3.3

First, we prove the lemma below.

**Lemma A.2.** *Let $\mathcal{F} : \mathbb{R}^d \times \mathbb{R}^d \to \mathbb{R}^d \times \mathbb{R}^d$ be measurable and let $\widetilde{\pi}$ be a measure on $\mathbb{R}^d \times \mathbb{R}^d$. Define $\pi := \mathcal{F}_\# \widetilde{\pi}$ (pushforward), i.e. $\pi(B) = \widetilde{\pi}(\mathcal{F}^{-1}(B))$ for all Borel sets $B$. Then for every Borel measurable $\psi : \mathbb{R}^d \times \mathbb{R}^d \to [0, \infty]$ (or integrable $\psi$), the standard change-of-variables formula for pushforwards gives*

$$\int \psi(z) \pi(dz) = \int \psi(\mathcal{F}(w)) \widetilde{\pi}(dw). \tag{A.4}$$

*Proof of Lemma A.2.* Let $(W, \mathcal{A})$ and $(Z, \mathcal{B})$ be measurable spaces, let $\mathcal{F} : W \to Z$ be measurable, and let $\widetilde{\pi}$ be a measure on $(W, \mathcal{A})$. Define the pushforward (image measure) $\pi := \mathcal{F}_\# \widetilde{\pi}$ on $(Z, \mathcal{B})$ by

$$\pi(B) := \widetilde{\pi}(\mathcal{F}^{-1}(B)) \ \forall B \in \mathcal{B}.$$

We prove that for every $\mathcal{B}$-measurable $\psi : Z \to [0, \infty]$ one has

$$\int_Z \psi(z) \pi(dz) = \int_W \psi(\mathcal{F}(w)) \widetilde{\pi}(dw). \tag{A.5}$$

(If $\psi$ is integrable, the same identity follows by applying the result to $\psi^+$ and $\psi^-$.)

**Step 1: indicator functions.** Let $\psi = \mathbf{1}_B$ for some $B \in \mathcal{B}$. Then, by definition of the integral of an indicator and of pushforward,

$$\int_Z \mathbf{1}_B(z) \pi(dz) = \pi(B) = \widetilde{\pi}(\mathcal{F}^{-1}(B)) = \int_W \mathbf{1}_{\mathcal{F}^{-1}(B)}(w) \widetilde{\pi}(dw)$$

$$= \int_W \mathbf{1}_B(\mathcal{F}(w)) \widetilde{\pi}(dw) = \int_W \psi(\mathcal{F}(w)) \widetilde{\pi}(dw).$$

**Step 2: nonnegative simple functions.** Let $\psi$ be simple: $\psi = \sum_{k=1}^n a_k \mathbf{1}_{B_k}$ with $a_k \geq 0$ and $B_k \in \mathcal{B}$. Using linearity of the integral and Step 1,

$$\int_Z \psi(z) \pi(dz) = \sum_{k=1}^n a_k \int_Z \mathbf{1}_{B_k}(z) \pi(dz) = \sum_{k=1}^n a_k \int_W \mathbf{1}_{B_k}(\mathcal{F}(w)) \widetilde{\pi}(dw)$$

$$= \int_W \Big( \sum_{k=1}^n a_k \mathbf{1}_{B_k}(\mathcal{F}(w)) \Big) \widetilde{\pi}(dw) = \int_W \psi(\mathcal{F}(w)) \widetilde{\pi}(dw).$$

**Step 3: general nonnegative measurable functions.** Let $\psi : Z \to [0, \infty]$ be $\mathcal{B}$-measurable. For each $m \in \mathbb{N}$, define the Borel (simple) function $\phi_m : [0, \infty] \to [0, \infty)$ by

$$\phi_m(t) := \begin{cases} 2^{-m} \lfloor 2^m t \rfloor, & 0 \leq t < m, \\ m, & t \in [m, \infty], \end{cases}$$

and set $\psi_m := \phi_m \circ \psi$. Then each $\psi_m$ is a nonnegative simple $\mathcal{B}$-measurable function, and

$$0 \leq \psi_1 \leq \psi_2 \leq \cdots \leq \psi \text{ and } \psi_m(z) \uparrow \psi(z) \ \forall z \in Z.$$

Applying Step 2 to each $\psi_m$ yields

$$\int_Z \psi_m d\pi = \int_W (\psi_m \circ \mathcal{F}) d\widetilde{\pi} \ \forall m.$$

Since $\psi_m \uparrow \psi$ implies $\psi_m \circ \mathcal{F} \uparrow \psi \circ \mathcal{F}$, the monotone convergence theorem gives

$$\int_Z \psi d\pi = \lim_{m \to \infty} \int_Z \psi_m d\pi = \lim_{m \to \infty} \int_W (\psi_m \circ \mathcal{F}) d\widetilde{\pi} = \int_W (\psi \circ \mathcal{F}) d\widetilde{\pi},$$

which proves (A.5) for all $\psi \geq 0$.

**Step 4: integrable (signed) functions.** If $\psi \in L^1(\pi)$ is real-valued, write $\psi = \psi^+ - \psi^-$ with $\psi^{\pm} \geq 0$ and $\int \psi^{\pm} d\pi < \infty$. Applying Step 3 to $\psi^{\pm}$ and subtracting gives

$$\int_Z \psi d\pi = \int_W (\psi \circ \mathcal{F}) d\widetilde{\pi}.$$

Moreover, applying Step 3 to $|\psi| = \psi^+ + \psi^-$ yields

$$\int_Z |\psi| d\pi = \int_W (|\psi| \circ \mathcal{F}) d\widetilde{\pi}.$$

Hence $\psi \in L^1(\pi)$ if and only if $\psi \circ \mathcal{F} \in L^1(\widetilde{\pi})$, and in that case the integrals coincide. $\qquad\square$

*Proof of Theorem 3.7.* **Step 1: Exact reduction to a self-transport QOT problem.** Define the (invertible) map

$$\mathcal{F} : \mathbb{R}^d \times \mathbb{R}^d \to \mathbb{R}^d \times \mathbb{R}^d, \mathcal{F}(x, v) := (x, T(v)).$$

Then $\mathcal{F}_{\#}(\mu \otimes \mu) = \mu \otimes \nu$. Hence, for any coupling $\widetilde{\pi} \in \Pi(\mu, \mu)$, $\pi := \mathcal{F}_{\#}\widetilde{\pi} \in \Pi(\mu, \nu)$ and:

$$\left\|\frac{d\pi}{d(\mu \otimes \nu)}\right\|_{L^2(\mu \otimes \nu)} = \left\|\frac{d\widetilde{\pi}}{d(\mu \otimes \mu)}\right\|_{L^2(\mu \otimes \mu)}.$$

Indeed, if $Q = \mathcal{F}_{\#}P$ and $\eta \ll P$, then $d(\mathcal{F}_{\#}\eta)/dQ = (d\eta/dP) \circ \mathcal{F}^{-1}$ and the $L^2$ norm is invariant under pushforward.

Because $T$ is affine and a gradient map, $\varphi$ is quadratic; more precisely, up to an additive constant,

$$\varphi(x) = \frac{1}{2}\langle x, Ax\rangle + \langle a, x\rangle, \varphi^*(y) = \frac{1}{2}\langle y - a, A^{-1}(y - a)\rangle.$$

Hence, the Fenchel-Young slack is explicit:

$$D_\varphi(x, T(v)) = \frac{1}{2}\langle v - x, A(v - x)\rangle.$$

By the standard slack identity (cf. Lemma 3.5 in this draft),

$$\int c d\pi - \mathrm{OT}(\mu, \nu) = \int D_\varphi d\pi.$$

In particular, using Lemma A.2 with $\psi = D_\varphi$ and $\mathcal{F}(x, v) = (x, T(v))$, we obtain

$$\int D_\varphi d\pi = \int D_\varphi(\mathcal{F}(x, v))\widetilde{\pi}(dx, dv)$$

$$= \int D_\varphi(x, T(v))\widetilde{\pi}(dx, dv).$$

Therefore, for $\pi = \mathcal{F}_{\#}\widetilde{\pi}$ we obtain

$$\int c d\pi - \mathrm{OT}(\mu, \nu) = \int D_\varphi(x, T(v))\widetilde{\pi}(dx, dv)$$

$$= \frac{1}{2}\int \langle v - x, A(v - x)\rangle\widetilde{\pi}(dx, dv).$$

Consequently, minimizing the original QOT objective over $\Pi(\mu, \nu)$ is equivalent (up to the additive constant $\mathrm{OT}(\mu, \nu)$) to minimizing over $\Pi(\mu, \mu)$ the self-transport functional

$$\widetilde{\pi} \mapsto \frac{1}{2}\int \langle v - x, A(v - x)\rangle d\widetilde{\pi}(x, v) + \frac{\varepsilon}{2}\left\|\frac{d\widetilde{\pi}}{d(\mu \otimes \mu)}\right\|_{L^2(\mu \otimes \mu)}^2. \tag{A.6}$$

In particular, if $\pi_\varepsilon$ is the QOT optimizer for $(\mu, \nu)$, then

$$\widetilde{\pi}_\varepsilon := (\mathrm{Id}, T^{-1})_\# \pi_\varepsilon$$

is the unique minimizer of (A.6) over $\Pi(\mu, \mu)$.

**Step 2: Sharp pointwise bias $\varepsilon^{1/(d+2)}$.** Define the linear change of variables

$$u := A^{1/2} x, w := A^{1/2} v,$$
$$\mu_A := (A^{1/2})_\# \mu, \widehat{\pi}_\varepsilon := (A^{1/2} \times A^{1/2})_\# \widetilde{\pi}_\varepsilon.$$

Then $\widehat{\pi}_\varepsilon \in \Pi(\mu_A, \mu_A)$ and (A.6) becomes the standard self-transport QOT problem

$$\widehat{\pi} \mapsto \int \frac{1}{2} \|u - w\|^2 d\widehat{\pi}(u, w) + \frac{\varepsilon}{2} \Big\| \frac{d\widehat{\pi}}{d(\mu_A \otimes \mu_A)} \Big\|^2_{L^2(\mu_A \otimes \mu_A)}.$$

Moreover, for every $(x, y) \in \mathrm{spt}\, \pi_\varepsilon$, letting $v = T^{-1}(y)$ gives $y - T(x) = T(v) - T(x) = A(v - x) = A^{1/2}(w - u)$, so

$$\|y - T(x)\| \le \|A^{1/2}\|_{\mathrm{op}} \|w - u\|.$$

Thus, a pointwise tube bound for $\widehat{\pi}_\varepsilon$ around the diagonal immediately yields a pointwise tube bound for $\pi_\varepsilon$ around $\mathrm{gr}\, T$.

We now invoke the sharp self-transport sparsity/tube theorem of Wiesel-Xu: in the self-transport case, the support satisfies

$$\sup_{(u,w) \in \mathrm{spt}\, \widehat{\pi}_\varepsilon} \|u - w\| \le C\big(\delta^{\mu_A}_{\mathrm{ST}}(\varepsilon)^{1/2} \wedge \mathrm{diam}(\mathrm{spt}\, \mu_A)\big),$$

where (for a measure $\eta$)

$$\delta^\eta_{\mathrm{ST}}(\varepsilon) := \inf\Big\{r > 0 : r\rho_\eta(\sqrt{r}) > \varepsilon\Big\},$$
$$\rho_\eta(s) := \inf_{z \in \mathrm{spt}\, \eta} \eta(B(z, s)).$$

See Wiesel & Xu, 2025, Theorem 3.1, 3.4. In particular, for all sufficiently small $\varepsilon$, the first term dominates, and we get

$$\sup_{(u,w) \in \mathrm{spt}\, \widehat{\pi}_\varepsilon} \|u - w\| \le C\delta^{\mu_A}_{\mathrm{ST}}(\varepsilon)^{1/2}, \tag{A.7}$$

In Wiesel & Xu, 2025, Lemma 3.5, Wiesel and Xu give $\|u - w\|^2 \le 4M$ on $\mathrm{spt}\, \pi_\varepsilon$ with $M := \sup_{u \in \mathrm{spt}\, \mu_A} f_\varepsilon(u)$, and their proof of Theorem 3.1 shows $M \le 16\delta_{\mathrm{ST}}(\varepsilon)$, hence $\|u - w\| \le 2\sqrt{M} \le 8\delta_{\mathrm{ST}}(\varepsilon)^{1/2}$. So, from (A.7) we have:

$$\sup_{(u,w) \in \mathrm{spt}\, \widehat{\pi}_\varepsilon} \|u - w\| \le 8\delta^{\mu_A}_{\mathrm{ST}}(\varepsilon)^{1/2}.$$

By the density lower bound $\underline{\lambda}_{\mu_A}$ on $\Omega_A$ and the definition of $\kappa_A$, for all $s \le r_A$ we have

$$\rho_{\mu_A}(s) \ge \underline{\lambda}_{\mu_A} \mathrm{Leb}(\Omega_A \cap B(u, s)) \ge \underline{\lambda}_{\mu_A} \kappa_A \omega_d s^d.$$

Hence for $r \le r_A^2$,

$$r\rho_{\mu_A}(\sqrt{r}) \ge \underline{\lambda}_{\mu_A} \kappa_A \omega_d r^{1+d/2}.$$

If $\varepsilon \le \varepsilon_0 = \underline{\lambda}_{\mu_A} \kappa_A \omega_d r_A^{d+2}$, then choosing

$$r_* := \Big(\frac{\varepsilon}{\underline{\lambda}_{\mu_A} \kappa_A \omega_d}\Big)^{\frac{2}{d+2}} \le r_A^2$$

gives $r_* \rho_{\mu_A}(\sqrt{r_*}) \ge \varepsilon$, hence $\delta^{\mu_A}_{\mathrm{ST}}(\varepsilon) \le r_*$ and therefore

$$\delta^{\mu_A}_{\mathrm{ST}}(\varepsilon)^{1/2} \le \Big(\frac{\varepsilon}{\underline{\lambda}_{\mu_A} \kappa_A \omega_d}\Big)^{\frac{1}{d+2}}.$$

Finally, we have

$$\|y - T(x)\| \le \|A^{1/2}\|_{\mathrm{op}} \|w - u\| = \sqrt{\lambda_{\max}(A)} \|w - u\|.$$

Combining with the bounds above yields (3.6). $\qquad\square$

*Proof of Proposition 3.8.* Write the (unnormalized) Gaussian densities

$$\phi_{m,\Sigma}(x) := (2\pi)^{-d/2} \det(\Sigma)^{-1/2} \exp\left(-\frac{1}{2}(x-m)^\top \Sigma^{-1}(x-m)\right).$$

Then $\mu$ has density $f_\mu(x) = Z_0^{-1}\phi_{m_0,\Sigma_0}(x)\mathbf{1}_{\Omega_0}(x)$ and $\nu$ has density $f_\nu(y) = Z_1^{-1}\phi_{m_1,\Sigma_1}(y)\mathbf{1}_{\Omega_1}(y)$.

Assume (3.9) and $T_\#\mu = \nu$.

Since $A$ is invertible, the standard change-of-variables formula yields for every $y \in \operatorname{spt}\nu$,

$$\begin{aligned} f_\nu(y) &= \frac{1}{\det A} f_\mu\big(A^{-1}(y-a)\big) \\ &= \frac{1}{\det A}\frac{1}{Z_0}\phi_{m_0,\Sigma_0}\big(A^{-1}(y-a)\big)\mathbf{1}_{\Omega_0}\big(A^{-1}(y-a)\big). \end{aligned} \tag{A.8}$$

Because $\phi_{m_0,\Sigma_0} > 0$ everywhere, the right-hand side is positive precisely on the set $A\Omega_0 + a$ up to Lebesgue-null sets, and vanishes outside it. Since $f_\nu > 0$ on $\Omega_1$ and $f_\nu = 0$ on $\mathbb{R}^d \setminus \Omega_1$, we must have

$$\Omega_1 = A\Omega_0 + a \text{ up to Lebesgue-null sets.}$$

On $\Omega_1$ (an open set), the indicators in (A.8) are equal to 1, so

$$\frac{1}{Z_1}\phi_{m_1,\Sigma_1}(y) = \frac{1}{\det A}\frac{1}{Z_0}\phi_{m_0,\Sigma_0}\big(A^{-1}(y-a)\big),\, y \in \Omega_1. \tag{A.9}$$

Taking logarithms and multiplying by $-2$, we obtain that there exists a constant $C \in \mathbb{R}$ such that for $y \in \Omega_1$:

$$\begin{aligned} &(y-m_1)^\top \Sigma_1^{-1}(y-m_1) \\ =&\big(A^{-1}(y-a) - m_0\big)^\top \Sigma_0^{-1}\big(A^{-1}(y-a)-m_0\big) + C, \end{aligned} \tag{A.10}$$

Both sides of (A.10) are quadratic polynomials in $y$; since $\Omega_1$ has nonempty interior, they agree as polynomials on $\mathbb{R}^d$. Writing

$$A^{-1}(y-a) - m_0 = A^{-1}\big(y - (Am_0 + a)\big),$$

we may rewrite the right-hand side of (A.10) as

$$\big(y - (Am_0 + a)\big)^\top \big(A^{-\top}\Sigma_0^{-1}A^{-1}\big)\big(y - (Am_0 + a)\big) + C.$$

Equality of the quadratic coefficients gives

$$\Sigma_1^{-1} = A^{-\top}\Sigma_0^{-1}A^{-1} \iff \Sigma_1 = A\Sigma_0 A,$$

and equality of the linear terms then forces $m_1 = Am_0 + a$. This proves the necessity of (3.10).

Conversely, assume (3.10). Then the density identity (A.9) holds (with the correct normalizing constants), hence $T_\#\mu = \nu$. Since $A = A^\top \succ 0$, the map $T(x) = Ax + a$ is the gradient of the convex function $\frac{1}{2}x^\top Ax + a^\top x$, and therefore it is the Monge map. $\qquad\square$

## B. Experiment details

### B.1. Deterministic solvers

**Non-linear Gauss-Seidel algorithm on** (B.6). Fix $g$ and update $f$ by solving for each $i$

$$\sum_{j=1}^{M} b_j[f_i + g_j - c_{ij}]_+ = \varepsilon.$$

---

**Algorithm 1** Non-linear Gauss-Seidel ((Lorenz et al., 2021, Alg. 1))

---

1: **Input:** weights $a \in \mathbb{R}_+^N$, $b \in \mathbb{R}_+^M$ with $\sum_i a_i = \sum_j b_j = 1$; cost matrix $C = (c_{ij})$; regularization parameter $\varepsilon > 0$; tolerance tol $> 0$; max iters $K$.
2: Initialize $g^{(0)} \in \mathbb{R}^M$ (e.g. $g^{(0)} = 0$), set $k \leftarrow 0$.
3: **repeat**
4:    **for** $i = 1$ **to** $N$ **do**
5:        Form $y_j^{(i)} \leftarrow c_{ij} - g_j^{(k)}$ for $j = 1, \dots, M$.
6:        Update $f_i^{(k+1)} \leftarrow$ SOLVEWEIGHTEDHINGE$(y^{(i)}, b, \varepsilon)$ using Algorithm 2.
7:    **end for**
8:    **for** $j = 1$ **to** $M$ **do**
9:        Form $z_i^{(j)} \leftarrow c_{ij} - f_i^{(k+1)}$ for $i = 1, \dots, N$.
10:       Update $g_j^{(k+1)} \leftarrow$ SOLVEWEIGHTEDHINGE$(z^{(j)}, a, \varepsilon)$ using Algorithm 2.
11:   **end for**
12:   (Gauge-fix) $\kappa \leftarrow \sum_{i=1}^N a_i f_i^{(k+1)}$; $f^{(k+1)} \leftarrow f^{(k+1)} - \kappa \mathbf{1}_N$, $g^{(k+1)} \leftarrow g^{(k+1)} + \kappa \mathbf{1}_M$.
13:   Compute residuals $r, s$ by (B.3).
14:   $k \leftarrow k + 1$.
15: **until** $\max_i |r_i| \vee \max_j |s_j| \leq$ tol **or** $k = K$
16: Recover $\pi$ via (B.5).
17: **Output:** $(f^{(k)}, g^{(k)}, \pi)$.

---

Let $y_{ij} := c_{ij} - g_j$; then this becomes the scalar monotone equation

$$\sum_{j=1}^M b_j (f_i - y_{ij})_+ = \varepsilon. \tag{B.1}$$

If $y_{i(1)} \leq \cdots \leq y_{i(M)}$ denotes the sorted list and $b_{(1)}, \dots, b_{(M)}$ are the corresponding permuted weights, define prefix sums

$$B_k := \sum_{\ell=1}^k b_{(\ell)}, S_k := \sum_{\ell=1}^k b_{(\ell)} y_{i(\ell)}.$$

On the interval $f_i \in [y_{i(k)}, y_{i(k+1)}]$, the left-hand side of (B.1) equals $B_k f_i - S_k$, hence the candidate solution is

$$f_i^{(k)} = \frac{\varepsilon + S_k}{B_k}.$$

The unique solution is obtained by finding $k$ such that $f_i^{(k)} \in [y_{i(k)}, y_{i(k+1)}]$ (with the convention $y_{i(M+1)} = +\infty$). The $g$-update is analogous: fixing $f$, solve for each $j$ the scalar equation $\sum_{i=1}^N a_i (g_j - (c_{ij} - f_i))_+ = \varepsilon$.

---

**Algorithm 2** SOLVEWEIGHTEDHINGE$(y, w, \varepsilon)$

---

1: **Input:** values $(y_j)_{j=1}^m \subset \mathbb{R}$, weights $(w_j)_{j=1}^m \subset (0, \infty)$, rhs $\varepsilon > 0$.
2: Sort $y_{(1)} \leq \cdots \leq y_{(m)}$ with permuted weights $w_{(1)}, \dots, w_{(m)}$; set $y_{(m+1)} \leftarrow +\infty$
3: Initialize $B \leftarrow 0$, $S \leftarrow 0$
4: **for** $k = 1$ **to** $m$ **do**
5:    $B \leftarrow B + w_{(k)}$, $S \leftarrow S + w_{(k)} y_{(k)}$
6:    $x \leftarrow (\varepsilon + S)/B$ {candidate root on $[y_{(k)}, y_{(k+1)}]$}
7:    **if** $y_{(k)} \leq x \leq y_{(k+1)}$ **then**
8:        **return** $x$
9:    **end if**
10: **end for**
11: **Output:** $x^\star \in \mathbb{R}$ solving $\sum_{j=1}^m w_j (x - y_j)_+ = \varepsilon$

---

---

**Algorithm 3** Globalized, regularized semismooth Newton for discrete QOT (cf. (Lorenz et al., 2021, Alg. 2))

---

1: **Input:** $a \in \mathbb{R}_+^N$, $b \in \mathbb{R}_+^M$; $C = (c_{ij})$; $\varepsilon > 0$; Newton regularization $\lambda > 0$; Armijo parameters $\theta, \xi \in (0, 1)$; tolerance $\text{tol} > 0$; max iters $K$.
2: Initialize $(f^{(0)}, g^{(0)})$ (e.g. output of Algorithm 1); set $k \leftarrow 0$.
3: **repeat**
4:  Compute slacks $P_{ij} \leftarrow f_i^{(k)} + g_j^{(k)} - c_{ij}$ and $Q_{ij} \leftarrow [P_{ij}]_+$.
5:  Active set $\sigma_{ij} \leftarrow \mathbf{1}_{\{P_{ij}>0\}}$.
6:  Form a Newton derivative $G$ as in (B.2).
7:  Solve the regularized linear system

$$(G + \lambda I)\begin{bmatrix} \Delta f \\ \Delta g \end{bmatrix} = -\begin{bmatrix} r \\ s \end{bmatrix}.$$

8:  Define the convex dual-minimization functional

$$\Phi_\varepsilon(f, g) := \frac{1}{2\varepsilon} \sum_{i=1}^N \sum_{j=1}^M a_i b_j [f_i + g_j - c_{ij}]_+^2 - \sum_{i=1}^N a_i f_i - \sum_{j=1}^M b_j g_j,$$

  (which satisfies $\max \text{dual (B.7)} = -\min \Phi_\varepsilon$).
9:  Set $t \leftarrow 1$ and compute the directional derivative

$$d \leftarrow \sum_{i,j} \pi_{ij}(\Delta f_i + \Delta g_j) - \sum_i a_i \Delta f_i - \sum_j b_j \Delta g_j, \quad \pi_{ij} := \frac{a_i b_j}{\varepsilon}[f_i^{(k)} + g_j^{(k)} - c_{ij}]_+.$$

10:  **while** $\Phi_\varepsilon(f^{(k)} + t\Delta f, g^{(k)} + t\Delta g) > \Phi_\varepsilon(f^{(k)}, g^{(k)}) + \theta t d$ **do**
11:   $t \leftarrow \xi t$.
12:  **end while**
13:  Update $(f^{(k+1)}, g^{(k+1)}) \leftarrow (f^{(k)}, g^{(k)}) + t(\Delta f, \Delta g)$.
14:  (Gauge-fix) $\kappa \leftarrow \sum_{i=1}^N a_i f_i^{(k+1)}$; $f^{(k+1)} \leftarrow f^{(k+1)} - \kappa \mathbf{1}_N$, $g^{(k+1)} \leftarrow g^{(k+1)} + \kappa \mathbf{1}_M$.
15:  Compute residuals $r, s$ by (B.3).
16:  $k \leftarrow k + 1$.
17: **until** $\max_i |r_i| \vee \max_j |s_j| \leq \text{tol}$ **or** $k = K$
18: Recover $\pi$ via (B.5).
19: **Output:** $(f^{(k)}, g^{(k)}, \pi)$.

---

**Semismooth Newton / quasi-Newton on the KKT system.** Define the residual vector $F(f, g) = (r, s)$ by (B.3). Introduce the activity matrix

$$\sigma_{ij} := \mathbf{1}_{\{f_i + g_j > c_{ij}\}}.$$

A Newton derivative of $F$ at $(f, g)$ is the block matrix

$$G = \begin{bmatrix} \text{diag}\left(\sum_{j=1}^M b_j \sigma_{ij}\right)_{i=1}^N & (b_j \sigma_{ij})_{i,j} \\ (a_i \sigma_{ij})_{j,i} & \text{diag}\left(\sum_{i=1}^N a_i \sigma_{ij}\right)_{j=1}^M \end{bmatrix}. \tag{B.2}$$

Because the potentials are shift-invariant,

$$(f, g) \mapsto (f + \kappa \mathbf{1}_N, g - \kappa \mathbf{1}_M) \Rightarrow [f_i + g_j - c_{ij}]_+ \text{ unchanged},$$

the residual map $F$ is constant along $(\mathbf{1}_N, -\mathbf{1}_M)$, and any Newton matrix has a nontrivial kernel (at least one-dimensional). Two standard remedies are: (i) fix a gauge, e.g. impose $\sum_{i=1}^N a_i f_i = 0$ (or set one coordinate to zero), so the reduced Newton matrix is invertible; or (ii) use a regularized step $(G + \lambda I)\Delta = -F$ with $\lambda > 0$.

**Stopping criterion.** Given $(f, g)$, define the residuals of (B.6)

$$r_i := \sum_{j=1}^{M} b_j [f_i + g_j - c_{ij}]_+ - \varepsilon, \quad s_j := \sum_{i=1}^{N} a_i [f_i + g_j - c_{ij}]_+ - \varepsilon, \tag{B.3}$$

and stop when $\max_i |r_i| \vee \max_j |s_j| \leq \text{tol}$, then form $\pi$ via (B.5).

## B.2. Details of synthetic data

**Compactly supported test families.** Choose a bounded Lipschitz domain $\Omega_0 \subset \mathbb{R}^d$ with nonempty interior and define

$$\Omega_1 := A\Omega_0 + a.$$

We use the compatible truncated Gaussian family: Fix $m_0 \in \mathbb{R}^d$ and $\Sigma_0 \in \mathbb{S}_{++}^d$, and define

$$m_1 := Am_0 + a, \Sigma_1 := A\Sigma_0 A.$$

Let $Z_0, Z_1 > 0$ be normalizing constants and set

$$
\begin{aligned}
d\mu(x) = & \frac{1}{Z_0} \exp\left( -\frac{1}{2}(x - m_0)^\top \Sigma_0^{-1} (x - m_0) \right) \\
& \cdot \mathbf{1}_{\Omega_0}(x) dx, \\
d\nu(y) = & \frac{1}{Z_1} \exp\left( -\frac{1}{2}(y - m_1)^\top \Sigma_1^{-1} (y - m_1) \right) \\
& \cdot \mathbf{1}_{\Omega_1}(y) dy.
\end{aligned}
$$

By Proposition 3.8, $\nu = T_{\#}\mu$ and $T$ is the Monge map. In this family, $T = \nabla\varphi$ is affine and $\nu = T_{\#}\mu$ by construction, hence $T$ is the Monge map.

**Finite-sample discretization.** We generate empirical measures as follows. Sample $x_1, \ldots, x_N \overset{\text{iid}}{\sim} \mu$. To form $\nu_M$, we use a partially paired pushforward: a fraction $p_{\text{pair}} \in (0, 1]$ of the $y_j$ are constructed as $y_j = T(x_j)$ (shared samples), and the remaining $1 - p_{\text{pair}}$ are constructed as $y_j = T(\tilde{x}_j)$ for independent $\tilde{x}_j \sim \mu$.

**Details of the synthetic instance family.** We use a diagonally structured $A$ with controlled conditioning: $A = \text{diag}(\text{base}, \text{base}^2, \ldots, \text{base}^d)$ with $\text{base} = 1.00005$ and $a = 0$. The truncation radius is set to $r_{\text{trunc}} = 0.8/d^{0.5}$, then we scale to ensure that $\mu$ and $\nu$'s supports are in $B(0, 1)$. For the truncated Gaussian, we use a weakly correlated covariance $\Sigma_{ii} = (\frac{1}{d} - \frac{45}{d^2})r_{\text{trunc}}^2$ and $\Sigma_{ij} = \frac{45}{d^2}r_{\text{trunc}}^2$ for $i \neq j$. The paired fraction is $p_{\text{pair}} = \min\{0.1, 0.1 \cdot (200/d)^2\}$.

## B.3. Discretization and the Discrete QOT Problem

Let $\mu_N = \sum_{i=1}^{N} a_i \delta_{x_i}$ and $\nu_M = \sum_{j=1}^{M} b_j \delta_{y_j}$ be discrete approximations of $\mu$ and $\nu$. In the affine pushforward setting $\nu = T_{\#}\mu$, it is convenient to sample $\nu$ by pushforward:

$$x_i \overset{\text{iid}}{\sim} \mu, \tilde{x}_j \overset{\text{iid}}{\sim} \mu, y_j := T(\tilde{x}_j) \sim \nu, a_i = \frac{1}{N}, b_j = \frac{1}{M}.$$

(One may also sample $y_j \sim \nu$ directly; this formula guarantees the exact affine relation at the population level.)

**Dual/KKT form (support-revealing).** Define the discrete quadratic cost matrix

$$c_{ij} := c(x_i, y_j) = \frac{1}{2} \|x_i - y_j\|^2, i = 1, \ldots, N, j = 1, \ldots, M.$$

The discrete quadratically regularized OT problem with reference weights $P_{ij} = a_i b_j$ is

$$\text{QOT}_\varepsilon(\mu_N, \nu_M) = \min_{\pi \in \Pi(a,b)} \left\{ \sum_{i=1}^{N} \sum_{j=1}^{M} c_{ij} \pi_{ij} + \frac{\varepsilon}{2} \sum_{i=1}^{N} \sum_{j=1}^{M} \frac{\pi_{ij}^2}{a_i b_j} \right\}, \tag{B.4}$$

where

$$\Pi(a,b) := \left\{ \pi \in \mathbb{R}_+^{N \times M} : \sum_{j=1}^{M} \pi_{ij} = a_i \; \forall i, \sum_{i=1}^{N} \pi_{ij} = b_j \; \forall j \right\}.$$

The discrete analogue of (2.2)-(2.4)-(2.5) is: there exist dual vectors $(f_i)_{i \leq N}$ and $(g_j)_{j \leq M}$ such that

$$\pi_{ij} = \frac{a_i b_j}{\varepsilon} \big[ f_i + g_j - c_{ij} \big]_+, \tag{B.5}$$

and the marginal constraints are equivalent to the nonlinear system

$$\sum_{j=1}^{M} b_j \big[ f_i + g_j - c_{ij} \big]_+ = \varepsilon \; \forall i, \sum_{i=1}^{N} a_i \big[ f_i + g_j - c_{ij} \big]_+ = \varepsilon \; \forall j. \tag{B.6}$$

Equations (B.6) admit efficient solvers (nonlinear Gauss-Seidel / coordinate descent and semismooth Newton methods) (Lorenz et al., 2021) and directly identify the discrete support $\{(i,j) : \pi_{ij} > 0\}$ via (B.5).

**Dual objective (discrete $\chi^2$ regularization).** For $\mu_N = \sum_{i=1}^{N} a_i \delta_{x_i}$ and $\nu_M = \sum_{j=1}^{M} b_j \delta_{y_j}$, the discrete dual associated with (1.1) is the unconstrained concave maximization

$$\max_{f \in \mathbb{R}^N, g \in \mathbb{R}^M} \left\{ \sum_{i=1}^{N} a_i f_i + \sum_{j=1}^{M} b_j g_j - \frac{1}{2\varepsilon} \sum_{i=1}^{N} \sum_{j=1}^{M} a_i b_j [f_i + g_j - c_{ij}]_+^2 \right\}. \tag{B.7}$$

The (everywhere-defined) gradient is

$$\nabla_{f_i}(\text{obj}) = a_i - \frac{a_i}{\varepsilon} \sum_{j=1}^{M} b_j [f_i + g_j - c_{ij}]_+, i = 1, \ldots, N,$$

$$\nabla_{g_j}(\text{obj}) = b_j - \frac{b_j}{\varepsilon} \sum_{i=1}^{N} a_i [f_i + g_j - c_{ij}]_+, j = 1, \ldots, M. \tag{B.8}$$

Thus the first-order optimality conditions $\nabla(\text{obj}) = 0$ are exactly the marginal system (B.6). In large-scale settings, (B.7) can be optimized by (stochastic) gradient ascent using mini-batches to estimate the sums in (B.8).

