# OpenReview forum: "Quadratically Regularized Optimal Transport: Localization Bounds and Affine Case Analysis"
_ICML.cc/2026/Conference — ICML 2026 regular_

### Official Review · Reviewer_eUcU · 2026-03-07

**Soundness:** 3
**Presentation:** 3
**Significance:** 3
**Originality:** 3
**Overall Recommendation:** 4
**Confidence:** 3

**Summary:**

This paper studies quadratically regularized optimal transport (QOT) with quadratic cost and asks how the QOT optimizer $\pi_\varepsilon$ localizes around the Monge coupling as $\varepsilon \downarrow 0$. The main result proves a general lower bound showing that the support of $\pi\varepsilon$ cannot concentrate around the Monge graph faster than order $\varepsilon^{1/(d+2)}$ in a directed Hausdorff/tube sense, matching the conjectured exponent under standard regularity assumptions. The paper also connects the QOT value gap to a mean-squared deviation from the Monge map under $\pi\varepsilon$, yielding an $\varepsilon^{2/(d+2)}$ scale and a high-probability tail bound. Finally, in an affine Brenier regime (including Gaussian-to-Gaussian transport), the paper derives a matching pointwise tube upper bound of order $\varepsilon^{1/(d+2)}$ and validates the predicted scaling via synthetic experiments across dimensions.

**Compliance With Llm Reviewing Policy:**

Affirmed.

**Key Questions For Authors:**

Key Questions for Authors.
1. Assumption sensitivity: How critical is the Lipschitz regularity of the Monge map for the lower bound exponent $1/(d+2)$? Can the argument be extended to weaker regularity (e.g., Hölder continuity of $T$) while retaining the same exponent, or does the exponent change?
2. Beyond compact support: Do you expect an analogous lower bound to hold for non-compact marginals such as untruncated Gaussians? If yes, what tail conditions would be sufficient to recover the same scaling (perhaps up to logarithmic factors)?
3. Regime of observability: For moderate discretization/sample sizes, can you characterize the $\varepsilon$ range where the asymptotic $\varepsilon^{1/(d+2)}$ regime should be visible, and how this depends on dimension $d$ and conditioning (e.g., affine map singular values in the Brenier case)?
4. General upper bounds: Are there intermediate regimes beyond affine $T$ (e.g., approximately affine / piecewise affine / small nonlinearity) where one can still obtain near-matching pointwise tube upper bounds? If so, what would be the main technical obstacle?

**Limitations:**

yes

**Strengths And Weaknesses:**

Strengths.
- The lower bound is clean and directly answers a natural question: it rules out faster-than-$\varepsilon^{1/(d+2)}$ localization of QOT to the Monge graph under standard density/spread and Lipschitz Monge map assumptions.
- The value-gap-to-bias link is conceptually useful: bounding the mean-squared deviation from the Monge map via $\Delta_\varepsilon$ provides an interpretable bridge from objective convergence to geometric control, and enables tail bounds when combined with known value-gap rates.
- The affine Brenier case is a strong complement.
- The experiments are well aligned with the theory.

Weaknesses.
- The general (non-affine) upper bound story is incomplete: the paper proves a sharp lower bound in full generality, but matching pointwise upper bounds are only shown in the affine regime, leaving a gap in the full localization characterization outside special cases.
- Several assumptions (compact support, density bounded above/below, Lipschitz Monge map) are standard but restrictive for some ML settings (e.g., non-compact Gaussians or heavy-tailed distributions).

---

> ### Author Rebuttal · Authors · 2026-03-30
>
> We thank the reviewer for the helpful feedback and thoughtful questions to improve our work. We will answer your questions as follows.
>
> For the first question, if $T$ is only $\alpha$-Hölder ($0<\alpha<1$), i.e.
> $$
> \\|T(x)-T(x')\\|\le L\\|x-x'\\|^\alpha,
> $$
> then from $\mathrm{dist}((x,y),\mathrm{gr}T)\le r$ we only get
> $$
> \\|y-T(x)\\|\le r + L r^\alpha \lesssim r^\alpha (r\ll 1),
> $$
> so the relevant $y$-ball radius becomes $\asymp r^\alpha$ and the small-ball volume is $\asymp r^{\alpha d}$. Re-running the fiberwise argument gives $\\|h\\|\_2^2\gtrsim r^{-\alpha d}$, hence
> $$
> \Delta_\varepsilon\gtrsim\varepsilon r^{-\alpha d}\Rightarrow r \gtrsim\varepsilon^{\frac{1}{\alpha(d+2)}}
> $$
> after combining with $\Delta_\varepsilon\lesssim \varepsilon^{2/(d+2)}$.
> Therefore the exponent deteriorates under weaker regularity: one cannot expect $1/(d+2)$ unless $\alpha=1$.
> We will add a remark stating this extension and giving the short derivation.
>
> For the second question, our lower-bound argument does not fundamentally require compact support: it uses (i) a uniform small-ball upper bound on $\nu$ and (ii) an upper bound on the value gap $\Delta_\varepsilon$. The quantization technique of Eckstein-Nutz [1] is explicitly designed for non-compact marginals and yields sharp value-gap rates under finite $(2+t)$-moments for quadratic cost. Therefore, for untruncated Gaussians (bounded density, all moments finite, and affine $T$), we expect the same $\varepsilon^{1/(d+2)}$ lower bound (with constants depending on $\overline\lambda_\nu$ and the Lipschitz constant of $T$).
>
> **More generally, a sufficient geometric condition is a uniform small-ball control of the form
> $$
> \sup_{y\in\mathbb R^d}\nu(B(y,r))\le 1\wedge (C r^d),
> $$
> combined with the moment assumptions needed to invoke the value-gap rate.**
> If only tail-dependent (non-uniform) small-ball bounds are available, one can recover the same exponent on high-probability truncations, potentially with additional logarithmic factors in $\varepsilon$ coming from the truncation radius.
>
> For the third question, we expected that we can usually see the $\varepsilon^{1/(d+2)}$ regime through relative error plotting, not depending on $d$ or other conditions, because we use regression to remove parameters that aren't $\varepsilon$. However, because of the curse of dimensionality, for larger $d$, the relative error may be farther from $0$.
>
> The reviewer can also check our answer to Reviewer McjX, Question 3.
>
> For the fourth question, we refer the reviewer to our answer to Reviewer oM9F, Question 3.
>
> [1] Eckstein, Stephan, and Marcel Nutz. "Convergence rates for regularized optimal transport via quantization." Mathematics of Operations Research 49.2 (2024): 1223-1240.

---

> > ### Author Rebuttal · Reviewer_eUcU · 2026-04-02
> >
> > Thank you for the response.

---

> > > ### Author Response · Authors · 2026-04-02
> > >
> > > Dear reviewer,
> > >
> > > We would like to sincerely thank you again for your feedback. We see that our rebuttal has fully resolved all of your remaining concerns about our work. If this was indeed the case, could you consider updating the evaluations, or at the very least explain your decision to keep the score at "weak accepted" accordingly?
> > >
> > > Best regards,
> > >
> > > The authors.

---

### Official Review · Reviewer_McjX · 2026-03-11

**Soundness:** 4
**Presentation:** 4
**Significance:** 4
**Originality:** 3
**Overall Recommendation:** 5
**Confidence:** 4

**Summary:**

The authors study the problem of quadratically regularized optimal transport (QOT), a variant of the optimal transport problem which adds a $L^2$ regularization term on the density of the transport plan. They focus on the problem of bounding the concentration of the support of the optimal QOT plan $\pi_\epsilon$ (where $\epsilon$ quantifies the strength of the regularization) towards the graph of the Monge map $T$ (the optimal map for (unregularized) optimal transport).

Under several assumptions on the source and target measures $\mu, \nu$ (compact support, bounded density, absolute continuity) and the Monge map $T$ (Lipschitz), the authors prove a lower bound on the vertical bias $\sup \|y-T(x)\|$ (where $(x,y)$ are taken in the support of $\pi_\epsilon$) in $\epsilon^{1/(d+2)}$, matching the one that was conjectured in the literature, as well as a tail bound on $\pi_\epsilon$. Moreover, in the case where the Monge map $T$ is affine, the authors prove a matching pointwise upper bound of $\|y-T(x)\|$ in $\epsilon^{1/(d+2)}$. Finally, the authors illustrate their theoretical results with numerical experiments on truncated Gaussians.

**Compliance With Llm Reviewing Policy:**

Affirmed.

**Final Justification:**

The authors' rebuttal adequately addressed my questions, so I confirm my positive score.

**Key Questions For Authors:**

1. Do you expect that the method used for proving the sharp pointwise upper bound in the affine Brenier regime could be adapted to the general regime, or would this require a completely different approach ?

2. In the experiments, you hypothesize that discretization floors, the support threshold and solver tolerances may bias the slope estimates upwards. Does increasing the values of M,N and decreasing the value of $\tau$ lead to an increase of the dimension $d^\ast$ where the estimated exponents cross the theoretically predicted one ?

3. The paper (and in particular Theorem 3.7) assumes that at least the source measure $\mu$ should be absolutely continuous, while the experiments are done on discretized measures. Are there any theoretical insights about how the discrete proxy $\widehat{bias}(\epsilon)$ converges to the actual vertical bias $b$ when $\tau \to 0$ and $M,N \to +\infty$ ?

4. Is the measure $\nu$ always absolutely continuous ? (Assumption 2 can be interpreted as bounding only the absolutely continuous part of $\nu$)

**Limitations:**

Yes.

**Strengths And Weaknesses:**

The paper is very well written, and all the proofs are rigorously carried out. Its findings are both novel and significant, and effectively fill a gap in the literature on the concentration of QOT plans, extending sharp bounds that were until now only known in very specific cases (such as self-transport or one-dimensional transport) to much more general settings where they were only conjectured (modulo some assumptions such as compact support or Lipschitz transport maps). Therefore, although some of the stronger results (such as the pointwise upper bound in Theorem 3.7) are only proved under the restrictive assumption that the Monge map is affine, in my opinion, the paper demonstrates more than sufficient merit to justify acceptance.

Some minor remarks:
- Lemma A.1 is standard (see for example Theorem 3.6.1 in Bogachev, *Measure theory*), and it probably isn't necessary to prove it
- Equation (A.9) is repeated twice (lines 763-769)

---

> ### Author Rebuttal · Authors · 2026-03-30
>
> We thank the reviewer for the helpful feedback and thoughtful questions to improve our work. We will answer your questions as follows.
>
> For the first question, we think this might not be a straightforward extension, because our results rely on the change of variables, which maps the Monge graph to the diagonal, turns the cost gap into a fixed quadratic form and reduces $\mathrm{QOT}(\mu,\nu)$ to a self-transport QOT problem where sharp tube results are available [1].
>
> In the non-affine case, one can still “straighten” the graph formally via $(x,y)\mapsto (x,T^{-1}(y))$, but the cost gap becomes a Bregman-type divergence tied to the Brenier potential, not a fixed quadratic form. Extending the self-transport sparsity machinery to this setting would likely require a new set of techniques (variable metric, nonlinearity, boundary effects).
>
> For the second question, yes, our expectation is that improving resolution (larger $N,M$) and reducing thresholding bias (smaller $\tau$) should push the onset of “floor-dominated” slope distortions to higher $d$, hence increase the empirical crossing dimension $d^\star$.
>
> For the qualitative mechanism:
> * For large $d$, the target exponent $1/(d+2)$ is extremely small, so the true bias changes only very weakly across the accessible $\varepsilon$ range; numerical floor can dominate the regression and distort slopes.
> * Lowering $\tau$ includes smaller-mass entries farther from the Monge graph, making the discrete proxy closer to the true support-based bias; increasing $N,M$ improves geometric resolution of the coupling.
>
> For the third question, in the following, we interpret the reviewer’s $\tau$ as our source measure $\mu$; in our main theorems, we assume $\mu\ll\mathrm{Leb}$.
>
> There are two logically separate issues:
>
> 1. Convergence of the regularized couplings under discretization / empirical approximation.
> There are now quantitative stability results showing that the optimizer varies Hölder-continuously (in Wasserstein distance) with perturbations of the marginals, and corresponding sample-complexity bounds when replacing $(\mu,\nu)$ by empirical measures [2].
> 2. Convergence of the support-based supremum functional $b_\varepsilon$.
> Even if $\pi^{N,M}\_\varepsilon\Rightarrow \pi_\varepsilon$ (or even if $W_p(\pi^{N,M}\_\varepsilon,\pi_\varepsilon)\to 0$), the map
> $$
> \pi \mapsto \sup_{(x,y)\in\mathrm{spt}\pi}\\|y-T(x)\\|
> $$
> is not continuous: an arbitrarily small amount of mass placed far from $\mathrm{gr} T$ changes the support and hence the supremum. For example,
> $$
> \pi_n:=\Big(1-\frac1n\Big)\pi_\varepsilon+\frac1nt_{(x_{\mathrm{far}},y_{\mathrm{far}})}
> $$
> satisfies $\pi_n\Rightarrow \pi_\varepsilon$ and $W_p(\pi_n,\pi_\varepsilon)\to 0$ for any fixed $p$, but $\sup_{\mathrm{spt}\pi_n}\\|y-T(x)\\|$ is dominated by the far atom.
> This discontinuity is exactly why our experiment reports a thresholded proxy
> $$
> \widehat{\mathrm{bias}}(\varepsilon)=\max_{(i,j):\pi^{N,M}\_{\varepsilon,ij}>\tau}\\|y_j-T(x_i)\\|.
> $$
> A rigorous convergence statement for $\widehat{\mathrm{bias}}(\varepsilon)$ to $b_\varepsilon$ would require an additional active-set stability / strict complementarity condition ensuring that there are no “vanishing-mass” far-away active edges that disappear in the limit.
> We do not currently have such a general theorem in $d>1$, and therefore we do not claim unconditional convergence of the support supremum from weak/Wasserstein convergence alone.
>
> For the fourth question, for our main geometric lower bound (and for the mean-square lower bound we added above), we use a uniform small-ball upper bound
> $$
> \nu(B(y,r))\le\overline\lambda_\nu\omega_dr^d,
> $$
> which is ensured by $\nu\ll\mathrm{Leb}$ (i.e., $\nu(dy)=\frac{d\nu}{dy}(y)dy$) with $\\|\frac{d\nu}{dy}\\|\_\infty\le \overline\lambda_\nu$.
> So yes, in Theorem 3.1’s regime, we are assuming $\nu$ is absolutely continuous.
> We will fix Assumption 2.
>
> [1] Wiesel, Johannes, and Xingyu Xu. "Sparsity of quadratically regularized optimal transport: Bounds on concentration and bias." SIAM Journal on Mathematical Analysis 57.6 (2025): 6498-6521.
>
> [2] Bayraktar, Erhan, Stephan Eckstein, and Xin Zhang. "Stability and sample complexity of divergence regularized optimal transport." Bernoulli 31.1 (2025): 213-239.

---

> > ### Author Rebuttal · Reviewer_McjX · 2026-04-03
> >
> > I thank the authors for their reply, which fully addresses my questions. I thus confirm my positive score.

---

### Official Review · Reviewer_oM9F · 2026-03-12

**Soundness:** 3
**Presentation:** 2
**Significance:** 2
**Originality:** 2
**Overall Recommendation:** 3
**Confidence:** 3

**Summary:**

This paper studies the convergence rate of quadratically regularized optimal transport plan to the true plan. In particular, the authors derive the lower bound showing that  the convergence rate with respect to the directed Hausdorff distance can not be faster than $\varepsilon^{1/(d+2)}$. They further show that this lower bound can be attained in the special case where the optimal transport map is affine. Using a different evaluation metric, i.e., the mean-squared derivation, they obtain a rate of $\varepsilon^{2/(d+2)}$ by applying an existing result on the QOT value gap.

**Compliance With Llm Reviewing Policy:**

Affirmed.

**Final Justification:**

I find the contribution of providing a lower bound to be limited. Moreover, the upper bound is weak, as it holds only for affine functions. Therefore, I believe this paper falls below the acceptance threshold.

**Key Questions For Authors:**

1, Is the bound with respect to mean-squared deviation optimal? Is there a lower bound w.r.t mean-squared deviation? Why are the convergence rates studied w.r.t sup metric (vertical bias)?

2, Assumption 1 may be vague, is $\mu$ assumed to be bounded from below on its support? If so, why do you introduce $\Omega = \mathrm{spt} \mu$.

3,  Could the paper be strengthened by providing a stronger upper bound beyond the affine map case? This may be achieved by assuming certain regularity and stability conditions for optimal transport.

**Limitations:**

The limitation that the upper bound is only derived for affine transport is stated in the paper.

**Strengths And Weaknesses:**

Soundness: the paper is technically sound. it provides a clear review of relevant literature and establishes a transparent connection to existing bounds. Some clarifications are needed for the high-probability mean-squared bound.

Presentation: the paper's presentation can be improved. Some constants are used inconsistently. For instance, b is introduced to represent the important vertical bias, but it is also used elsewhere as a generic constant with different meaning. Some notations are used before they are formally introduced in section 2, for instance gr T and spt pi appear already in introduction. Finally, in my opinion, the related work section would be better placed before the main results.

Significance: the paper addresses an important problem. The results advance our understanding about localization rate of QOT optimizers to the Monge graph. The matching upper bound is weak as it only applies to affine transport. This is the main reason I assign a rating of 3 to this paper.

Originality: the theoretical results are original. The new lower bound could be influential.

---

> ### Author Rebuttal · Authors · 2026-03-30
>
> We thank the reviewer for the helpful feedback and thoughtful questions to improve our work. We will answer your questions as follows, and hope that it will help resolve the reviewer's concerns about our work.
>
> For the first question, the reviewer is right to ask for a lower bound for the mean-squared deviation. In fact, we can strengthen the paper with a matching lower bound (same exponent) under the current assumption, with an addition to Assumption 2 (used in Lemma 3.2): $\nu\ll\mathrm{Leb}$.
> Let
> $$
> m_\varepsilon:=\int \\|y-T(x)\\|^2\pi_\varepsilon(dx,dy)\ ,\quad t:=\sqrt{2m_\varepsilon}.
> $$
> **Lemma (new, lower bound for mean-squared deviation).**
> Assume $\nu$ is absolutely continuous with density $d\nu/dy\in L^\infty$ and $\\|\frac{d\nu}{dy}\\|\_\infty\le\overline\lambda_\nu$.
> Then for all $y,r,\nu(B(y,r))\le 1\wedge(\overline\lambda_\nu\omega_dr^d)$.
> Hence, for all sufficiently small $\varepsilon$,
> $$
> m_\varepsilon\ge c(d,\overline\lambda_\nu,C_{\mathrm{val}})\varepsilon^{2/(d+2)}.
> $$
> **Proof sketch**: define $\pi_\varepsilon=h(\mu\otimes\nu)$, so $\int h(x,y)\mu(dx)=1$ for $\nu$-a.e. $y$.
> Let $t=\sqrt{2m_\varepsilon}$. Using Markov inequality, we have $$\pi_\varepsilon(\\|y-T(x)\\|\le t)\ge1/2.$$
> This implies that there is a set $D\subset\mathrm{spt}\nu$ with $\nu(D)\ge 1/3$ such that $\int_{T^{-1}(B(y,t))}h(x,y)\mu(dx)\ge 1/4$ for $y\in D$.
> Using Cauchy-Schwarz inequality and the fact that $\nu(B(y,t))\le1\wedge(\overline\lambda_\nu\omega_dt^d)$, we have
> $$\int h(x,y)^2\mu(dx)\gtrsim t^{-d}\text{ for }y\in D.$$
> Hence $\\|h\\|\_{L^2(\mu\otimes\nu)}^2\gtrsim m_\varepsilon^{-d/2}$.
> Since $\Delta_\varepsilon\ge\frac{\varepsilon}{2}\\|h\\|\_{L^2}^2$, we obtain $\Delta_\varepsilon\gtrsim\varepsilon m_\varepsilon^{-d/2}$ and conclude $m_\varepsilon\gtrsim\varepsilon^{2/(d+2)}$.
> Thus, together with our existing upper bound $m_\varepsilon\lesssim\varepsilon^{2/(d+2)}$, the mean-squared deviation is $\Theta(\varepsilon^{2/(d+2)})$ in exponent under our standing regularity regime.
>
> We will add this lemma (with explicit constants) to directly answer the optimality question.
>
> **Why study $\sup$-type bias (vertical bias) at all?**
> We still study the vertical bias $b_\varepsilon$ because $m_\varepsilon$ is an average control under $\pi_\varepsilon$ and does not rule out rare but large deviations that determine support geometry (sparse graphs/correspondences).
>
> For the second question, yes, we assume $\mu$ has a density bounded from below and above on its support.
> We introduce $\Omega$ to make $\mu$'s condition explicit and to avoid ambiguities about the boundary of $\mathrm{spt}\mu$.
>
> Concretely, we will rewrite Assumption 1 to this form:
> There exists a bounded connected Lipschitz domain $\Omega\subset\mathbb{R}^d$ and a density $\rho_\mu$ such that
> $$
> \mu(dx)=\rho_\mu(x)\mathbf{1}_\Omega(x)dx,0<\underline\lambda\_\mu\le\rho\_\mu(x)\le\overline\lambda\_\mu<\infty\text{ for a.e. }x\in\Omega.
> $$
> Then $\mathrm{spt}\mu=\overline{\Omega}$ automatically. This eliminates the perceived redundancy.
>
> For the third question, we agree with the reviewer that this is the main missing piece. Our current sharp pointwise upper bound leverages a global straightening of the graph into a diagonal via an affine change of variables, which is special to affine $T$.
>
> We see two plausible future directions for a stronger setting beyond affine (we will mention them more explicitly):
>
> 1. **Bi-Lipschitz / uniformly elliptic Brenier regime.** If one assumes uniform bounds
> $$
> 0\prec mI\preceq\nabla T(x)\preceq LI\text{ (a.e.)}
> $$
> and sufficient regularity (e.g., $T$ is $C^{1,1}$), then the Fenchel-Young slack becomes a Bregman divergence with a uniformly bounded Hessian.
> This suggests a route via anisotropic self-transport estimates, but it would require extending sparsity/tube arguments (Minty-type detachment and density estimates) to Bregman costs rather than the Euclidean quadratic cost.
>
> 2. **Approximately affine / small nonlinearity.** If $T$ is close to affine on scales comparable to the conjectured tube radius $r\sim\varepsilon^{1/(d+2)}$ (e.g., $\\|\nabla^2 T\\|_\infty r \ll 1$), then the Bregman cost can be viewed as a perturbation of a constant quadratic form.
> The main obstacle is that QOT support is defined by strict inequalities in the dual slack; small perturbations can create or destroy thin “active regions,” so one needs quantitative stability of the active set (a strict-complementarity type condition).
>
> We view these as substantial but plausible extensions. We will extend the “Limitations/Open Questions” to reflect these concrete pathways.

---

> > ### Author Rebuttal · Reviewer_oM9F · 2026-04-02
> >
> > I thank the authors for pointing out potential extensions based on Bi-Lipschitz conditions. However, the current evidence is limited, and it is not clear that such extensions are feasible without substantial additional work. I therefore maintain my rating.

---

> > > ### Author Response · Authors · 2026-04-03
> > >
> > > Dear Reviewer oM9F,
> > >
> > > We appreciate the reviewer's engagement with our rebuttal. We want to clarify a point about the scope of our contributions that may not have been fully conveyed.
> > >
> > > We agree that extending the sharp pointwise upper bound beyond the affine Brenier regime would be an important next step. At the same time, we respectfully think this should be viewed as a substantial open problem rather than a missing ingredient for the present submission.
> > >
> > > The main contribution of the paper is broader than Theorem 3.7 alone. Concretely, the paper provides:
> > >
> > > i) the first general lower bound at the conjectured scale $\varepsilon^{\frac{1}{d+2}}$ under standard assumptions;
> > >
> > > ii) a general value-gap-to-bias inequality yielding the $\varepsilon^{\frac{1}{d+2}}$ mean-square/high-probability scale; and
> > >
> > > iii) a matching pointwise upper bound in the affine Brenier regime.
> > >
> > > To summarize, latest works about solving the $\varepsilon^\frac{1}{d+2}$ rate conjecture:
> > > 1. Wiesel-Xu [1] solved the conjecture, only in the self-transport case ($\mu=\nu$).
> > > 2. González-Sanz-Nutz [2] solved the conjecture, only in the $d=1$ case.
> > > 3. Gvalani-Koch [3] gave a better bound for the general case, but it is still $\varepsilon^{O(1/d^2)}$.
> > >
> > > We also want to clarify that the affine result isn't just a sanity check. Its proof uses a structural reduction: a global affine straightening of $\mathrm{gr}T$ onto the diagonal, which reduces the problem to a self-transport QOT problem. This mechanism is specific to the affine case and does not appear to extend routinely to nonlinear $T$. For this reason, obtaining a matching sharp upper bound in the fully non-affine setting would require genuinely new ideas, rather than a small strengthening of the current argument. We will revise the manuscript to make this scope even more explicit, and to emphasize more clearly that the fully non-affine sharp upper bound is future work, not a claim of the present paper.
> > >
> > > Regarding the feasibility of extensions suggested in our rebuttal: we proposed the bi-Lipschitz/uniformly elliptic and approximately affine directions as concrete pathways, not as *established results*. We are happy to tone down the language and move them to "Open Questions" if the reviewer felt they were oversold. Our intent was to show that natural intermediate steps exist between the affine regime and the fully general case.
> > >
> > > **In summary, our view is that the remaining issue concerns the scope of a natural next problem, rather than the novelty or significance of the results established here. We maintain the opinion that requiring its resolution as a condition for acceptance would be out of scope for the current 8-page limitation of our submission.**
> > >
> > >
> > > Best regards,
> > >
> > > The authors.
> > >
> > > [1]: Wiesel, Johannes, and Xingyu Xu. "Sparsity of quadratically regularized optimal transport: Bounds on concentration and bias." SIAM Journal on Mathematical Analysis 57.6 (2025): 6498-6521.
> > >
> > > [2]: González-Sanz, Alberto, and Marcel Nutz. "Sparsity of quadratically regularized optimal transport: Scalar case." arXiv preprint arXiv:2410.03353 (2024).
> > >
> > > [3]: Gvalani, Rishabh S., and Lukas Koch. "Sparsity and uniform regularity for regularised optimal transport." arXiv preprint arXiv:2601.05130 (2026).

---

### Official Review · Reviewer_JgPx · 2026-03-13

**Soundness:** 3
**Presentation:** 4
**Significance:** 3
**Originality:** 3
**Overall Recommendation:** 5
**Confidence:** 3

**Summary:**

This paper studies quadratically regularized optimal transport (QOT) and, more specifically, the localization of the QOT optimizer  $\pi_\varepsilon$ around the graph of the Monge map T as $\varepsilon \to 0$.

**Compliance With Llm Reviewing Policy:**

Affirmed.

**Key Questions For Authors:**

Could the authors comment on the extent to which these findings might extend beyond \( L_2 \) regularization? I would be particularly interested in the authors' perspective on whether analogous localization or bias-control results might hold for **expectile-based regularization**, for instance in the spirit of ENOT (ENOT: Expectile Regularization for Fast and Accurate Training of Neural Optimal Transport, Buzun et al., 2024). A brief discussion in the  Related Work section even if speculative would help connect the paper's theoretical contributions to modern regularization techniques in related domains like neural optimal transport.

**Limitations:**

yes

**Strengths And Weaknesses:**

Strengths

1) The paper focuses on a well-motivated and theoretically meaningful problem.

2) The lower-bound result, if fully correct as stated, would be a notable contribution to the theory of QOT.

3) The connection between the value gap and the map-level bias is elegant and useful.

4) The paper is generally well structured, and the narrative of the results is easy to follow.

Weaknesses

1) The argument in Theorem 3.7 appears plausible, but I would appreciate a more explicit treatment of:
how supports transform under the pushforward and affine change of variables?
why the geometric constant $\kappa_A$ is strictly positive under the stated assumptions?

2) The experiments are consistent with the theory but remain fairly narrow in scope, as they only address a synthetic affine regime and mainly serve as a sanity check.

---

> ### Author Rebuttal · Authors · 2026-03-30
>
> We would like to thank the reviewer for pointing out an interesting reference that might have a connection with our work. For the first question, our lower-bound mechanism is a geometric “tube vs. density blow-up” argument using only: (i) marginal constraints, (ii) a small-ball upper bound for $\nu$, and (iii) an upper bound on the regularized value gap.
> Quadratic structure enters only through “regularizer", or the norm of $h=d\pi/d(\mu\otimes\nu)$.
>
> This extends directly to $L^p$ regularization ($p>1$):
> $$
> \mathrm{OT}\_{p,\varepsilon}(\mu,\nu) := \left\\{\inf\_{\pi\in\Pi(\mu,\nu)}\int cd\pi+\frac{\varepsilon}{p}\Big\\|\frac{d\pi}{dP}\Big\\|\_{L^p(P)}^p\right\\}, P=\mu\otimes\nu.
> $$
>
> The dual/KKT still has a *hinge* structure: the dual penalty becomes $\int[f+g-c]\_+^qdP$ with $q=\frac{p}{p-1}$ and the optimal density is
> $$
> h(x,y)=\Big(\frac{[f(x)+g(y)-c(x,y)]\_+}{\varepsilon}\Big)^{\frac{1}{p-1}},
> $$
> so exact sparsity of $\mathrm{spt}\pi_{\varepsilon,p}$ persists.
>
> **Lower bound.** Replacing Cauchy-Schwarz by Hölder yields $\\|h\\|\_p^p\gtrsim r^{-d(p-1)}$ when $\mathrm{spt}\pi_{\varepsilon,p}$ lies in an $r$-tube around $\mathrm{gr} T$, hence the regularizer costs at least $\varepsilon r^{-d(p-1)}$.
> Combining with the sharp quadratic-cost value-gap bounds for power-type regularizers $f(t)=t^\rho/\rho$ [1, Thm. 3.8 and Prop 4.4] yields $\mathrm{OT}^{(p)}\_\varepsilon-\mathrm{OT}=\Theta\left(\varepsilon^{2/(d(p-1)+2)}\right)$ and hence $\mathrm{dist}(\mathrm{spt}\pi_{\varepsilon,p};\mathrm{gr}T)\gtrsim\varepsilon^{1/(d(p-1)+2)}$.
>
> **Upper bounds.** Two points:
> 1. Our mean-squared bias inequality is regularizer-agnostic: for any $\pi\in\Pi(\mu,\nu)$,
> $$
> \int\\|y-T(x)\\|^2d\pi\le2L\Big(\int cd\pi-\mathrm{OT}(\mu,\nu)\Big),
> $$
> so for $\pi_{\varepsilon,p}$ we get $\mathbb{E}\_{\pi_{\varepsilon,p}}\\|Y-T(X)\\|^2\le2L(\mathrm{OT}\_{p,\varepsilon}-\mathrm{OT})=O(\varepsilon^{\frac{2}{d(p-1)+2}})$.
>
> 2. For pointwise tube bounds ($\sup_{(x,y)\in\mathrm{spt}\pi_{\varepsilon,p}}\\|y-T(x)\\|$), the only sharp self-transport results we currently leverage are proved for the quadratic ($p=2$) penalty.
> Extending Wiesel-Xu’s self-transport tube analysis to the $[\cdot]\_+^q$ dual penalty ($q\neq2$) looks feasible but nontrivial (active-set geometry and detachment estimates must be redone), and we will flag this explicitly as an open direction.
>
> For the second question, ENOT [2] introduces expectile regularization of the dual Kantorovich potentials to stabilize neural OT training, in particular enforcing binding conditions for the $c$-transform step; it is therefore a dual/potential-side regularization rather than a primal $f$-divergence penalty on couplings w.r.t. $\mu\otimes\nu$. So, support sparsity and tube localization of $\mathrm{spt}\pi_\varepsilon$ are not directly implied by ENOT’s results.
>
> That being said, there are two natural bridges we will articulate in the Related Work section in the extended version of the manuscript:
>
> 1. If an “expectile divergence” is used as an $f$-divergence regularizer on $\pi$ (primal side), then because the penalty is still quadratic growth around the reference (power-type), we expect the same localization exponent as $L^2$ (up to constants depending on the asymmetry parameter), by the same tube-vs.-norm argument.
> 2. ENOT’s asymmetric squared-loss viewpoint highlights a broader design axis: penalizing dual infeasibility with different weights (positive vs. negative slack). Our hinge-squared dual already corresponds to the extreme asymmetric choice “penalize only positive slack.” ENOT suggests that intermediate asymmetries can improve optimization without introducing notable bias in certain regimes.
>
> We will add a brief paragraph summarizing ENOT’s mechanism and clarifying that our geometric sparsity results pertain to coupling regularization (primal $f$-divergence / Orlicz penalties), while ENOT is primarily a neural training regularization of the conjugate transform.
>
> [1] Eckstein, Stephan, and Marcel Nutz. "Convergence rates for regularized optimal transport via quantization." Mathematics of Operations Research 49.2 (2024): 1223-1240.
>
> [2] Buzun, Nazar, Maksim Bobrin, and Dmitry V. Dylov. "Expectile regularization for fast and accurate training of neural optimal transport." Advances in Neural Information Processing Systems 37 (2024): 119811-119837.

---

> > ### Author Rebuttal · Reviewer_JgPx · 2026-04-03
> >
> > Thank you for the detailed rebuttal.
> >
> > The authors addressed my questions. In particular, the rebuttal clarified the possible extension to power-type regularization and ENOT-style dual regularization. I also appreciate that the technical details around Theorem 3.7 will be made more explicit in the revision.

---

### Decision · Program_Chairs · 2026-04-30

**Decision:**

Accept (regular)

**Comment:**

This paper studies quadratically regularized optimal transport (QOT) and addresses the problem that how fast the support of the regularized optimizer localizes around the Monge graph as the regularization parameter $\varepsilon \to 0$. The main result is a general lower bound showing that localization cannot occur faster than order $\varepsilon^{1/(d+2)}$ in directed Hausdorff distance under standard regularity assumptions. The paper also establishes a value-gap-to-bias relation yielding an $\varepsilon^{2/(d+2)}$ mean-squared deviation scale, and proves a matching sharp pointwise upper bound in the affine Brenier regime, including Gaussian-to-Gaussian transport.

The reviewers generally agreed that the paper is technically strong, clearly written, and addresses a meaningful theoretical problem. Overall, the review profile was clearly positive, with two Accepts, one Weak Accept, and one Weak Reject. The main concern was about scope, clarifying assumptions, notation, and the interpretation of the mean-squared bias result. These are fair limitations, but not outweighing the paper’s main contribution, as it fills a real gap in the current understanding of QOT localization.

For the final version, we encourage the authors to make the scope of the affine upper bound especially explicit, and to incorporate the rebuttal clarifications on assumptions, the mean-squared lower bound, and the broader connections to related regularization schemes.